# Model-based local density sharpening of cryo-EM maps

**Arjen J Jakobi[1,2,3], Matthias Wilmanns[2], Carsten Sachse[1]***

[1]Structural and Computational Biology, European Molecular Biology Laboratory, Heidelberg, Germany; [2]Hamburg Unit c/o DESY, European Molecular Biology Laboratory, Hamburg, Germany; [3]The Hamburg Centre for Ultrafast Imaging, Hamburg, Germany

**Abstract** Atomic models based on high-resolution density maps are the ultimate result of the cryo-EM structure determination process. Here, we introduce a general procedure for local sharpening of cryo-EM density maps based on prior knowledge of an atomic reference structure. The procedure optimizes contrast of cryo-EM densities by amplitude scaling against the radially averaged local falloff estimated from a windowed reference model. By testing the procedure using six cryo-EM structures of TRPV1, β-galactosidase, γ-secretase, ribosome-EF-Tu complex, 20S proteasome and RNA polymerase III, we illustrate how local sharpening can increase interpretability of density maps in particular in cases of resolution variation and facilitates model building and atomic model refinement.
DOI: https://doi.org/10.7554/eLife.27131.001

**\*For correspondence:**
carsten.sachse@embl.de

**Competing interests:** The authors declare that no competing interests exist.

## Introduction

Electron cryo-microscopy (cryo-EM) has been used as a method to visualize biological macromolecules in their native-hydrated state for more than three decades (*Adrian et al., 1984*). Major improvements in detector technology (*Faruqi and Henderson, 2007*; *McMullan et al., 2016*) and associated computational procedures recently transformed single-particle cryo-EM that since has been producing a plenitude of near-atomic resolution structures from specimens of lower symmetry (*Allegretti et al., 2014*; *Amunts et al., 2014*; *Bai et al., 2013*) and lower molecular weight than previously deemed possible (*Bai et al., 2015*; *Liao et al., 2013*; *Merk et al., 2016*). At this resolution, the reconstructed EM density maps contain sufficient detail to interpret the structure using atomic models. Atomic models are frequently used by biologists without expert knowledge to interrogate function based on mechanistic hypotheses inferred from the structure. The building and refinement of atomic models thus present a critical step in the structure determination process and their accuracy depends on the quality of the EM density.

Sophisticated protocols for map interpretation as well as parameterization and validation of coordinate refinement in X-ray crystallography exist and these procedures have been adapted to work with cryo-EM maps (*Amunts et al., 2014*; *Brown et al., 2015*; *Fromm et al., 2015*; *Hoffmann et al., 2015*; *Wang et al., 2014*). Coordinate refinement in X-ray crystallography typically depends on initial phase estimates that are iteratively improved (*Agarwal and Isaacs, 1977*; *Grosse-Kunstleve et al., 2002*; *Lunin et al., 2002*; *Read, 1986*; *Wilson, 1942*). As a result, crystallographic refinement successively improves map interpretability throughout the refinement procedure. By contrast, the cryo-EM experiment yields both amplitudes and phases directly from images and experimentally determined EM density can therefore be used as a constant minimization target (*Grigorieff et al., 1996*; *Unwin, 2005*; *Yonekura et al., 2003*).

Regardless of implementation, all model building and refinement procedures are guided by the structural information content present in the data and primarily depend on resolution. The

commonly accepted procedure for resolution estimation in cryo-EM is the Fourier shell correlation (FSC), which measures the correlation of Fourier coefficients in resolution shells between independent map reconstructions (*Harauz and van Heel, 1986*; *Saxton and Baumeister, 1982*; *van Heel et al., 1982*). The reported map resolution corresponds to a pre-defined cut-off of the FSC curve (*Rosenthal and Henderson, 2003*). Currently, it is common practice to low-pass filter cryo-EM volumes at this resolution cut-off. Many of the recently determined EM density maps, however, contain regions that differ substantially from this overall resolution due to varying flexibility and/or subunit occupancy in different parts of the structure (*Bai et al., 2013*; *Hoffmann et al., 2015*; *Zhao et al., 2015*). In addition to inherent flexibility of macromolecular complexes, the quality of the reconstruction is poorer at the particle periphery when compared with the particle center due to limitations of current three-dimensional (3D) reconstruction algorithms. In providing an estimate of overall map resolution, the average FSC measurement falls short of describing such local resolution differences in different parts of the structure (*Cardone et al., 2013*).

In addition to resolution, density contrast is critical for map interpretation. High-resolution contrast in cryo-EM maps is attenuated by a resolution-dependent amplitude falloff. Compensation is achieved by sharpening with a uniform map B-factor, which partially restores contrast loss in cryo-EM maps (*Rosenthal and Henderson, 2003*) and in low-resolution electron density maps from X-ray crystallography (*DeLaBarre and Brunger, 2003*; *Jacobson et al., 1961*; *Stehle and Harrison, 1996*). To avoid noise amplification at high resolution, a signal-to-noise weighting scheme based on the FSC equivalent to the crystallographic figure-of-merit (FOM) is commonly used (*Rosenthal and Henderson, 2003*). The B-factor can be estimated numerically (*DeLaBarre and Brunger, 2003*; *Rosenthal and Henderson, 2003*), but in practice often needs to be fine-tuned empirically in an effort to generate optimal contrast. Alternative scaling procedures use the X-ray solution scattering (SAXS) curve of the specimen (*Gabashvili et al., 2000*; *Schmid et al., 1999*) or the radially averaged amplitude profile of the crystal structure (*Falke et al., 2005*) to correct the amplitudes accordingly. The desired outcome of all procedures is to enhance contrast such that it reveals expected molecular features at the given resolution while avoiding undue amplification of background noise. Depending on the resolution, such features comprise clearly defined signatures of secondary structure and amino acid side chains. With the wealth of recent near-atomic resolution cryo-EM reconstructions (*Cheng, 2015*; *KühlbrandtKuhlbrandt, 2014*; *Subramaniam et al., 2016*), building and refinement of atomic models have become the ultimate steps of cryo-EM structure determination. Therefore, new approaches are required to optimize the process of sharpening by objective means and to better integrate model refinement with map interpretation.

Crystallographic model refinement routinely uses prior information from atomic models to enhance the clarity of electron density maps. As the model improves, better phase estimates iteratively make the electron density more interpretable and ultimately provide more accurate structures. Current cryo-EM density interpretation does not make use of model information and model refinement strategies leave the target density unchanged. We here present a general procedure that utilizes prior model information to iteratively compute map representations better suited for interpretation and atomic model refinement than those obtained from existing approaches. The procedure optimizes map contrast by local amplitude scaling against an atomic structure and thereby implicitly accounts for differences in local resolution. Based on several test cases from the EMDB model challenge (http://challenges.emdatabank.org), we show that model-based local sharpening can facilitate model building and refinement by increasing local density contrast.

## Results

### Improved image contrast by local reference-based amplitude scaling

The appearance of map features does not only depend on the presence of amplitude and phase signal at the corresponding resolution, but also on the relative magnitude of these frequency components. This is particularly true for cryo-EM images that are affected by amplitude modulations due to the contrast-transfer function (*Erickson and Klug, 1970*). To examine the effect of various amplitude-based contrast enhancing procedures, we manipulated the amplitudes of a perfect two-dimensional (2D) test image of 512 × 512 pixel dimensions (*Figure 1A*). For illustration purposes, we substituted the amplitudes of the image with random Gaussian noise and computed the inverse

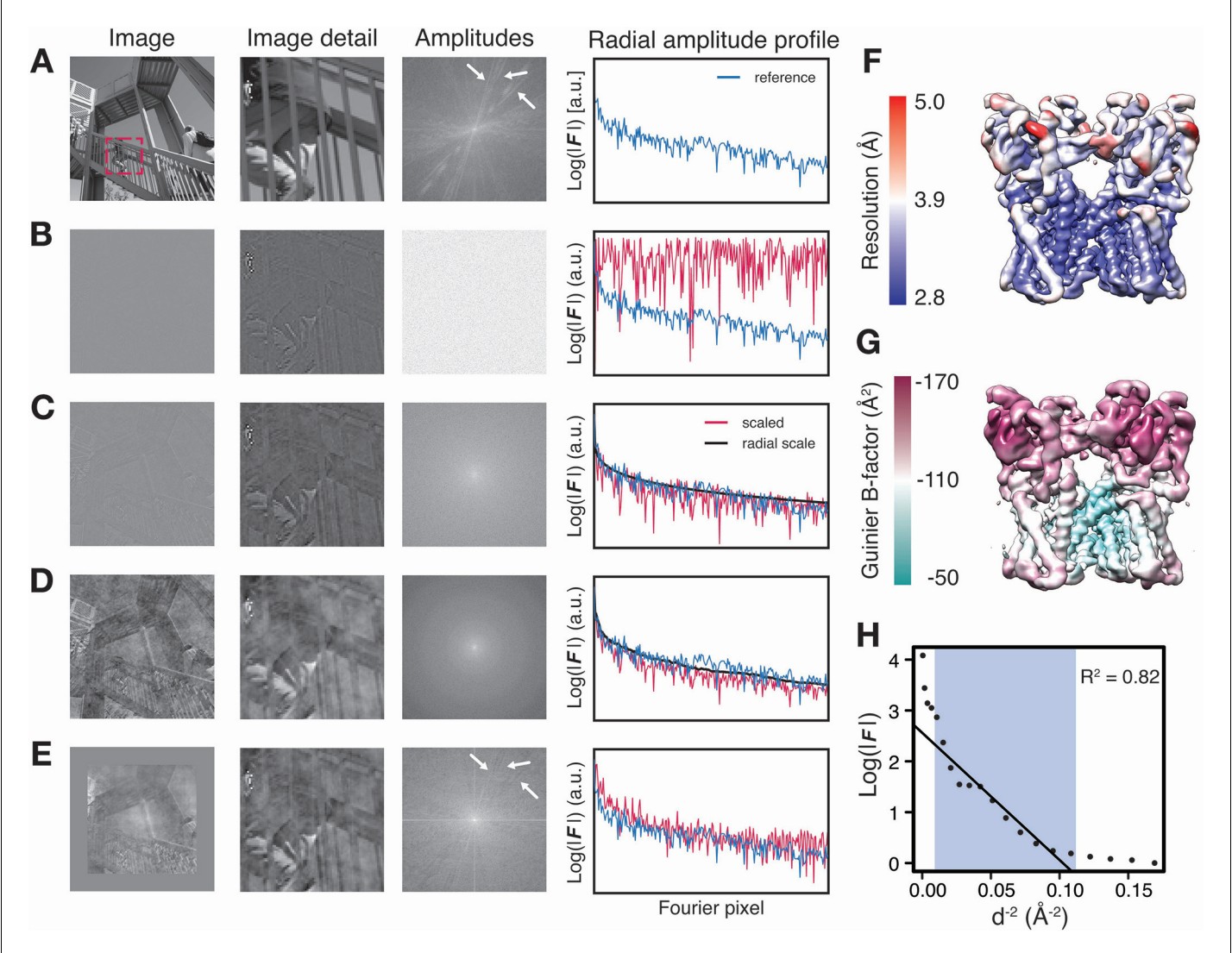

**Figure 1.** Illustration of local amplitude scaling (LocScale) using a 2D test image and determination local B-factors in cryo-EM maps. (**A**) The original image (left), a magnified image detail (center left), the overall amplitude spectrum (center right) and the radially averaged amplitude profile (right) are displayed in panels along columns, respectively. Applied procedures are shown along rows. The radial amplitude profile of the reference, the scaled image and the applied scale are shown in blue, red and black, respectively. (**B**) Hybrid image resulting when amplitudes were replaced by random Gaussian noise and combined with phases from the original image in (**A**). The falloff of the amplitude profile of the hybrid image (red) is flat over the entire frequency range and hampers visual perception of image contrast. (**C**) Hybrid image from (**B**) when random amplitudes were scaled by a negative exponential estimated from the amplitude falloff of the original image. (**D**) Hybrid image from (**C**) with random amplitudes scaled by the average radial falloff profile from (**A**). (**E**) Hybrid image from (**D**) with amplitudes scaled locally to reference amplitudes from (**A**) computed in tiles of 128 × 128 pixel rolling windows. Note that diagonal lines in the amplitude spectrum (white arrows) arising from periodic image features are optimally recovered. Compare with the power spectrum of the original image. (**F**) Resolution estimates from local FSC calculation with a 24 Å sampling window mapped onto the surface representation of the TRPV1 channel (EMD-5778). (**G**) B-factors determined from local Guinier fitting in 24 Å sampling window mapped onto the surface representation of the TRPV1 channel (EMD-5778). (**H**) Guinier plot for a representative density window used for local B-factor estimation in (**G**). The linear falloff estimated from least squares fitting to the data is shown as straight line. The fitting interval ranging from 10 Å to the locally estimated resolution (here 3.1 Å) is highlighted in blue.

DOI: https://doi.org/10.7554/eLife.27131.002

The following figure supplement is available for figure 1:

**Figure supplement 1.** Local B-factor estimation by Guinier fitting.

DOI: https://doi.org/10.7554/eLife.27131.003

Fourier transform of the hybrid image. Although this image still has perfect phases and the overall image information is largely conserved, it shows poor contrast and thus makes feature discrimination difficult, and in part impossible (*Figure 1B*). As reflected in the radial amplitude profile of the original image, image amplitudes decay exponentially towards higher frequencies. Amplitudes of cryo-EM maps decay much faster due to imaging imperfections (*Henderson, 1992*). Scaling amplitudes by an approximated exponential falloff restores the overall magnitude distribution at low and high frequencies and improves image contrast (*Figure 1C*). This procedure is equivalent to the widely used cryo-EM map sharpening procedure with a numerically estimated B-factor. Better contrast can be obtained by more accurately restoring amplitude differences between adjacent frequency components. This can be achieved by scaling amplitudes against the radial average of the original image, equivalent to so-called reference-based scaling of cryo-EM maps using an X-ray model or solution scattering profile (*Figure 1D*). Moreover, as the radially averaged amplitude profile sums many features across the image, the relative distribution of adjacent frequency components may not be optimally restored by global amplitude correction. We therefore tested whether local image scaling can yield more accurate amplitude distributions. To this end, we implemented a tile-based scaling procedure, in which the amplitudes of 128 × 128 pixel rolling windows were scaled against the local radial average of the respective reference window and the central pixels of the scaled windows make up the resulting image. In this image, high-resolution contrast including directional amplitude components is better restored when compared with that scaled against the global average, and the power spectrum more closely resembles that of the original image (*Figure 1E*). This 2D example highlights the importance of relative amplitude scaling for local image contrast and shows that contrast restoration by tile-based amplitude scaling against a local reference should be preferred over compensating amplitude decay by an exponential or an overall falloff approximation.

## Local amplitude scaling optimizes contrast in cryo-EM maps

Due to the contrast benefit of reference-based local amplitude scaling in photographic images, we set out to assess whether this strategy also improves contrast restoration in cryo-EM density maps. Contrary to the 2D image with uniform resolution, cryo-EM maps display local resolution variations in addition to the steep amplitude decay towards high resolution (*Figure 1F*). We quantified this amplitude decay and mapped locally determined B-factors to four deposited 3D densities from the EM databank (*Figure 1G*, *Figure 1–figure supplement 1A–C*). We chose three cryo-EM structures from the 2015/2016 EMDB model challenge (http://challenges.emdatabank.org), TRPV1 channel (EMD-5778), γ-secretase (EMD-3061) and β-galactosidase (EMD-2984), and one additional test case from our own lab, Pol III (EMD-3180). For this analysis, we performed least squares fitting of the exponential amplitude falloff for overlapping density windows within resolution ranges better than 10 Å up to the locally determined resolution cutoff (*Figure 1H*). It should be noted that estimating B-factors by linear fitting to Guinier profiles is error-prone with coefficients of determination $R^2$ ranging from 0.6 to 0.9, in particular in areas of poorer resolution (*Figure 1—figure supplement 1A,D and E*). The results from our four test cases suggest that locally adjusted sharpening levels may be required to optimally represent density features, in particular for maps with significant local resolution variation.

In order to account for local differences in the estimated B-factor, we implemented three different local sharpening procedures and compared them with the commonly used global sharpening approach. First, we employed local sharpening using locally determined Guinier B-factors and FOM weighting. Second, we tested the unsharp masking technique (UNSHARP), which has previously been reported to improve interpretability of crystallographic maps (*Afonine et al., 2015*). In a third procedure, we locally scaled (LocScale) the experimental amplitudes to match the radially averaged amplitude profile of a map generated from a refined atomic reference model (*Figure 2A*) in a 3D procedure analogous to the tile-based approach for the 2D test image described above. Within a rolling density window, this procedure adapts the relative amplitude levels required to optimize local contrast by imposing the expected radially averaged amplitude profile for the structure contained within it, and by acting as a convoluted Gaussian low-pass filter that reduces background noise (*Figure 2B*). All techniques, including global B-factor sharpening, considerably enhance interpretability of the density beyond that of EMD-5778 (*Figure 2C*). The comparison of the results obtained with the three local sharpening procedures reveals that UNSHARP and LocScale maps contain lower levels of peripheral noise compared with maps obtained by sharpening using locally estimated

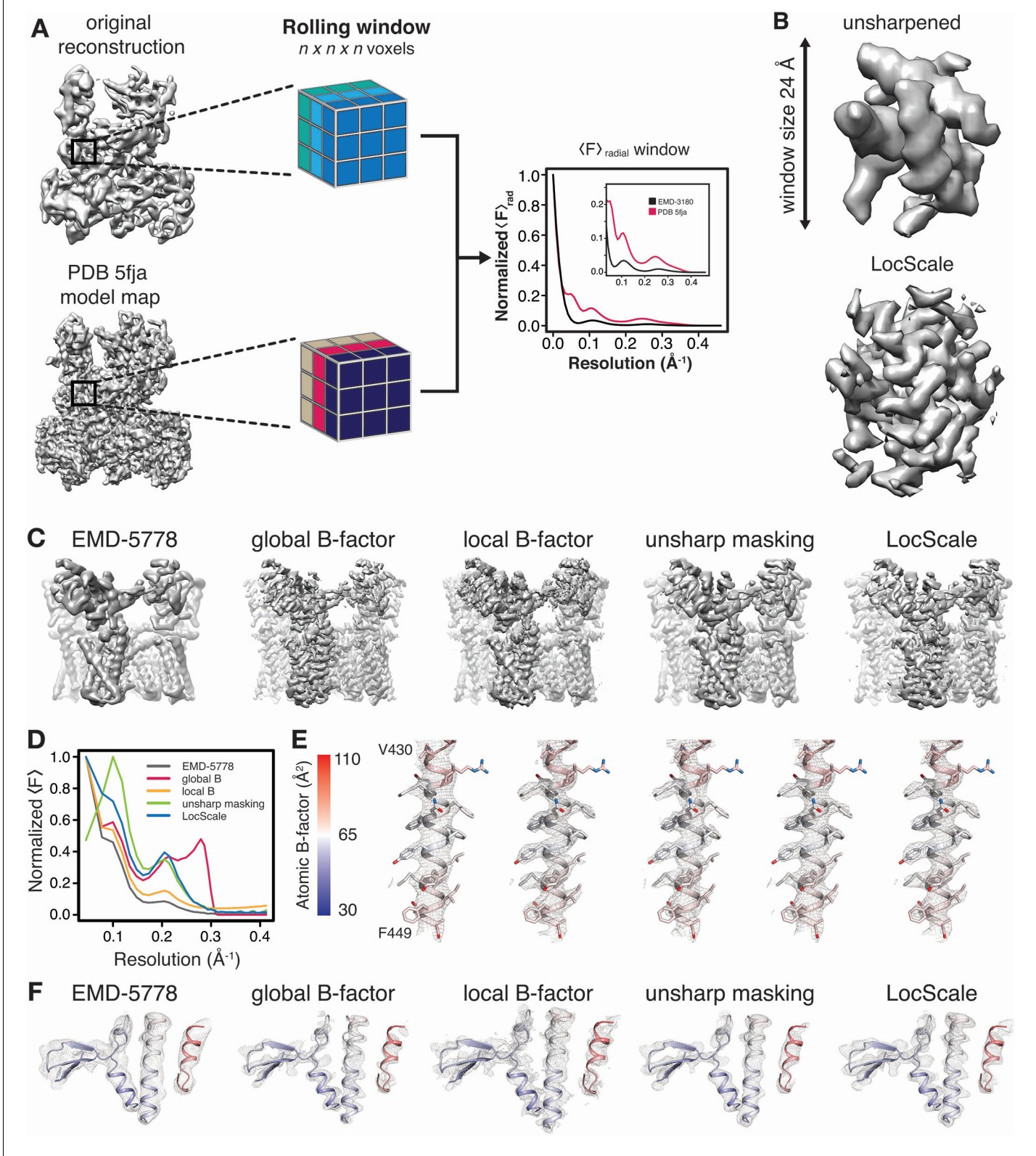

**Figure 2.** Generation of 3D LocScale density maps and comparison of global and local sharpening procedures. (**A**) Schematic illustration of the LocScale procedure for a Pol III density map. Two equivalent map segments (rolling windows) are extracted from the original 3D reconstruction of EMD-3180 (top) and a map simulated from the atomic model (PDB ID 5fja) (bottom). For each rolling window, the radially averaged structure factor profile is computed and used to scale the amplitudes of the corresponding window of the original reconstruction. Note, the fine structure of the model

*Figure 2 continued on next page*

*Figure 2 continued*

amplitudes has a characteristic profile in the resolution range <10 Å due to protein secondary structure and deviates from a simple exponential falloff (inset). (B) Effect of amplitude scaling illustrated for an exemplary density window. The density contained within a window cube is shown before (unsharpened) and after (LocScale) application of amplitude scaling. The central voxel of the rolling window is assigned the map value after amplitude scaling. The procedure is repeated by moving the window along the map until each voxel has been assigned a density value based on the locally estimated contrast. (C) Side views of TRPV1 densities obtained with different sharpening methods. Far left, EMD-5778 unsharpened; left, global Guinier B-factor −100 Å$^2$; middle, local Guinier B-factor; right, unsharp masking and far right, LocScale sharpening. (D) Radial amplitude profiles of the respective maps shown in (C). (E) Mesh representation of the densities for transmembrane residues 430–449 superposed on the atomic model (color-coded by atomic B-factor). The order from left to right is the same as in (C) and (F). Side chains are shown in stick representation. (F) Mesh representation for a peripheral density region superposed on the atomic model shown in ribbon representation and color-coded by atomic B-factor. B-factor scale as in (E).

DOI: https://doi.org/10.7554/eLife.27131.004

The following figure supplements are available for figure 2:

**Figure supplement 1.** Effect of secondary structure and atomic B-factors on the radial structure factor amplitude profile.

DOI: https://doi.org/10.7554/eLife.27131.005

**Figure supplement 2.** Comparison of global and local sharpening methods for Pol III, γ-secretase and β-galactosidase.

DOI: https://doi.org/10.7554/eLife.27131.006

Guinier B-factors despite the FOM weighting. As expected, the overall radial amplitude profiles of the respective maps show falloffs that deviate from an exponential decay of randomly distributed atoms in the 7 to 4 Å range (*Figure 2D*). This deviation, also termed Debye effect, is mainly due to characteristic distances of protein secondary structure (*Figure 2—figure supplement 1A–C*) (*Morris et al., 2004*). Comparison of the profiles further suggests that global sharpening can lead to overrepresentation of high frequency components, whereas the LocScale map displays the expected smooth falloff towards high frequencies. This behavior arises from the local distribution of atomic B-factors that determines the steepness of the amplitude falloff and thus implicitly also accounts for local resolution differences (*Figure 2—figure supplement 1D–H*). Unlike the other sharpening methods, UNSHARP masking appears to result in attenuation of low frequencies (*Figure 2D*). Inspection of well-defined map regions (e.g. residues 430–449) indicates that all sharpening approaches can be adjusted locally such that they reveal a high level of interpretable detail (*Figure 2E*). Numerical B-factor based approaches, however, show poorer quality density in particular in peripheral map regions resolved at lower than average resolution. While reproducing density comparably sharp in well-resolved regions, density from UNSHARP masking appears more blurred in less well-resolved regions when compared with the LocScale map (*Figure 2F*). Similar observations can be made for density maps of Pol III, β-galactosidase and γ-secretase (*Figure 2—figure supplement 2A–D*). In conclusion, reference-based sharpening by local amplitude scaling provides a robust way to appropriately sharpen and filter all regions in a map simultaneously such that they conform to the expected amplitude falloff.

## LocScale maps improve interpretability of cryo-EM maps

We illustrate the results of the LocScale procedure for cryo-EM maps of Pol III (EMD-3180) and TRPV1 channel (EMD-5778) in comparison with the currently most widely used procedure of uniform sharpening. Overall, the LocScale map of Pol III appears cleaner than the uniformly sharpened map and secondary structure elements are readily recognizable (*Figure 3A*). At first sight, the Pol III LocScale map appears less featured at the peripheral subunits when compared with EMD-3180. However, closer inspection reveals that features in the deposited map in fact are sharpening artifacts inconsistent with molecular structure (*Figure 3A*, upper panel). By contrast, LocScale sharpening eliminates such artifacts resulting from incorrect amplitude scaling (*Figure 3A*, lower panel). In a second example, we compared the LocScale map of the TRPV1 channel with the deposited EMD-5778 map. This comparison reveals more clearly resolved α-helical pitch and well-defined side chain densities in the LocScale map (*Figure 3B*, lower panel), which are not as readily discernible in the uniformly sharpened and filtered deposited map (*Figure 3B*, upper panel). In effect, the optimized contrast improves visibility of high-resolution features from map regions at higher than average resolution in addition to dampening noise in regions at lower than average resolution. Local sharpening helps increase interpretability of cryo-EM density maps by optimally restoring local map contrast.

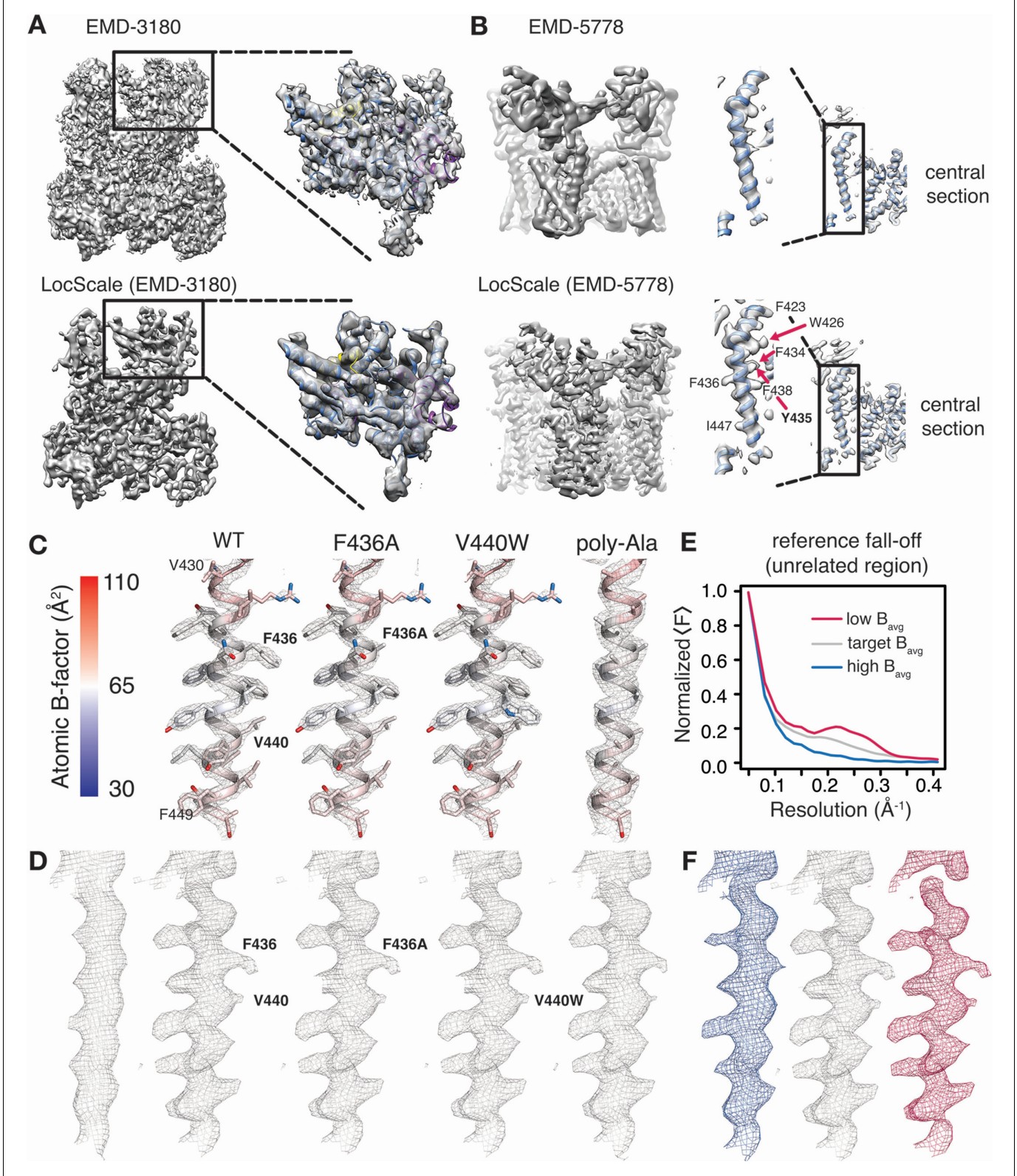

**Figure 3.** Optimized cryo-EM map contrast by local model-based amplitude scaling (LocScale) and assessment of model bias. (A) and (B) Illustration of the LocScale sharpening for initially over-sharpened (A), RNA Pol III; EMD-3180) and under-sharpened maps (B,) TRPV1; EMD-5778). The maps after application of global (Guinier) sharpening (top) are shown in comparison with its LocScale equivalent (bottom). The LocScale map contains recognizable secondary structure. Zoomed-in inset: (A) The globally sharpened map shows undue noise amplification rather than true molecular features of the

*Figure 3 continued on next page*

*Figure 3 continued*

peripheral subunits as visualized in the LocScale map. Central section: (**B**) When global sharpening levels were underestimated, LocScale map reveals previously invisible features such as defined side chain densities (arrows). (**C**) Mesh representation of simulated reference densities for TRPV1 residues 430–449 for wild-type (WT), F436A, V440W and poly-alanine mutants superposed on the respective atomic model. Atoms are colored by atomic B-factor with B-factor scale as defined on the far left. (**D**) Mesh representation of density for residues 430–449 as deposited in EMD-5778 (far left) and LocScale maps obtained by sharpening using WT, F436A, V440W and poly-alanine mutants as a reference maps. The order follows that in (**C**). (**E**) Radial amplitude profile of a structurally unrelated region used as reference window for scaling with high (blue), target (grey) and low (red) average B-factors. The respective regions are shown in *Figure 3—figure supplement 1*. The denotation high and low refers to the average B-factor across the reference window compared with that estimated for the target window. (**F**) Densities for residues V420-F449 obtained after scaling using the reference falloffs in (**E**) with equivalent color-coding.

DOI: https://doi.org/10.7554/eLife.27131.007

The following figure supplements are available for figure 3:

**Figure supplement 1.** Assessment of coordinate and B-factor perturbation on LocScale densities and the relationship of atomic B-factors to local resolution.

DOI: https://doi.org/10.7554/eLife.27131.008

**Figure supplement 2.** Effect of atomic B-factors on the LocScale procedure.

DOI: https://doi.org/10.7554/eLife.27131.009

**Figure supplement 3.** Effect of window size on local reference-based sharpening.

DOI: https://doi.org/10.7554/eLife.27131.010

## Atomic model coordinates do not introduce feature bias

Model bias is commonly understood as the appearance or disappearance of map features not present in the experimental data, but imposed by the model. Model bias is a serious concern in X-ray crystallography, where phase information from the model is combined with experimentally determined Fourier amplitudes to compute crystallographic electron density maps. We note here that throughout the LocScale procedure no phase information is transferred from the model to the scaled density maps. However, information from model amplitudes radially averaged over a finite-sized density window is used to correct the power of Fourier amplitudes in the experimental frequency spectrum. To ascertain that map features observed in the LocScale maps originate from enhanced contrast and not from model bias, we examined a selected transmembrane helix (residues 430–449) of the TRPV1 structure after computing multiple LocScale maps using radial amplitudes from different reference structures (*Figure 3C–D*). In a first experiment, we substituted F436 to alanine and used the F436A mutant to compute a reference map. Despite the absence of side chain density in the reference map, side chain density for F436 is clearly visible in the scaled map. In a second experiment, we replaced V440 with tryptophan and used the V440W mutant to compute a reference map. In this case, no additional density is visible in the scaled map despite W440 density being present in the reference map. Finally, we truncated all side chains in the model and produced a poly-alanine model as a scaling reference. This procedure will substantially decrease the overall scattering mass of the reference. While side chain densities are slightly attenuated when compared at equivalent density threshold, they are readily interpretable despite the absence of any side chain information in the reference model. In order to further assess the potential of coordinate bias, we randomly perturbed atom positions of the model within a mask encompassing the molecular outline and computed reference maps from the perturbed models to scale the experimental density. For perturbations ranging from 2.5 to 50 Å r.m.s.d., side chain densities are readily observed despite the reference having no resemblance to the original structure. We do, however, observe a noticeable decrease in map contrast with increasing r.m.s.d (*Figure 3—figure supplement 1A*). Together, these results of mutating individual side chains and strongly perturbing model coordinates confirm that observed density features indeed are the result of enhanced contrast and not of coordinate bias from the reference model.

## Atomic B-factors are required for optimal sharpening

Coordinate perturbations of the reference did not result in appearance or disappearance of features in the sharpened map, whereas coordinate randomization beyond a certain r.m.s.d. resulted in suboptimal sharpening. Coordinate randomization will change local distributions and local correlations of B-factors across the reference map and thus will alter the falloff of the radial structure factor

computed across the scaling windows. Similarly, LocScale maps computed from references that do not accurately model the effective variation of local B-factors (and resolution) across the experimental density will therefore also result in suboptimal sharpening. To quantify the effect of atom randomization on the sharpening result, we computed kurtosis for LocScale maps obtained from scaling against coordinate-perturbed reference models. (*Figure 3—figure supplement 1B*). Kurtosis has been previously suggested as a measure for sharpness, or 'peakedness', of crystallographic density maps (*Afonine et al., 2015*). Effectively, as model perturbation increases, local B-factor correlations are gradually lost until converging to a uniform average B-factor across the volume contained within the mask (*Figure 3—figure supplement 1C*). This way, local sharpening slowly converges to match the result obtained from global sharpening with a B-factor determined from the overall amplitude falloff of the experimental density map.

From the above, it follows that optimal performance of the LocScale procedure is insensitive to the underlying coordinate model as long as the local B-factor is correctly approximated by the radially averaged amplitude falloff across the reference windows. To further illustrate this effect, we performed an experiment in which we used a structurally unrelated reference window from the center of the molecule to scale the target density window at the molecule's periphery comprising the transmembrane helix formed by residues V430-F449 (*Figure 3—figure supplement 1D*). Based on the atomic B-factor distribution across the TRPV1 molecule, the B-factors of atoms contained within the reference window are lower than those contained within the target window. Consequently, the radial amplitude falloff of the reference window is shallower than the actual local amplitude profile expected for the target window (*Figure 3E*) and, while still representing the features of the structure accurately, the scaled density starts to show signs of density fragmentation due to moderate oversharpening (*Figure 3F*, magenta). Next, we artificially inflated the B-factor of the reference window and repeated the procedure. The radial amplitude falloff of the reference window is now steeper than the expected falloff and the resulting density is blurred (*Figure 3E–F*, blue; *Figure 3—figure supplement 1D–F*). Last, we imposed a reference falloff with a B-factor similar to that estimated from the average B-factor of atoms contained in the target window, resulting in improved sharpening of comparable quality with that obtained with the 'true' reference window at the target location (*Figure 3E–F*, gray; *Figure 3—figure supplement 1D–F*). This example confirms our previous conclusion that a structurally unrelated window can be used for scaling, while it also illustrates how the result of sharpening is dependent on the correct estimation of the slope of the amplitude falloff in the reference window, which in the LocScale procedure is determined from the refined atomic B-factors of the atomic model.

## Atomic B-factors model local resolution in cryo-EM maps

If refinement proceeds robustly, atomic B-factors are expected to correlate with local resolution estimates, as exemplified for Pol III (EMD-3180 and PDB ID 5fja; *Figure 3—figure supplement 1G,H*). To further examine the effect of estimated B-factors on the scaling procedure, we generated LocScale maps of EMD-3180 based on reference maps from models whose atoms were assigned a series of uniform B-factors ranging from 0 to 300 $Å^2$ (*Figure 3—figure supplement 2A*). Where atomic B-factors were underestimated (B $\leq$ 50 $Å^2$) for model subunits located in the low-resolution map region at the particle periphery, the shallow falloff in the scaling reference led to density fragmentation in the scaled map (*Figure 3—figure supplement 2B–D*). Conversely, a local amplitude profile that falls off too steeply by overestimated atomic B-factors within the core of the molecule (B $\geq$ 200 $Å^2$) resulted in blurring of well resolved secondary structure elements in the respective scaled map (*Figure 3—figure supplement 2B,C*). Therefore, in line with our results from reference window randomization, incorrectly estimated atomic B-factors can lead to a suboptimal representation of density features. For correctly estimated atomic B-factor distributions, the level of blurring and sharpening is locally adapted for each region leading to optimized density contrast in the LocScale map (*Figure 3—figure supplement 2A–D*).

## The effect of window size on the scaled densities

The LocScale procedure employs scaling of radial amplitude profiles that are averaged over a rolling density window spanning dimensions between 15 and 45 Å in the described examples. Window size is a parameter that may have important consequences for the performance of local sharpening. We

therefore tested the effect of the window size parameter on our selected maps by systematically increasing the size of the scaling window used in the sharpening procedure. For our experiment, we used the TRPV1 map along with an incomplete, but refined reference model (PDB ID 3j9j) covering the transmembrane region of the TRPV1 channel structure. The smallest practically applicable window size corresponds to eight pixels, as sampling with less than four Fourier pixels is too low for meaningful rotational averaging. It is instructive to analyze the Guinier plots of the resulting LocScale maps (*Figure 3—figure supplement 3A*). First, the absolute amplitude at low frequency increases with larger window sizes. Larger windows will cover more density at the periphery of the model and will lead to scaling larger portions of adjacent experimental density not covered by the reference, resulting in an increased absolute amplitude of the scaled map. Second, windows of 14 pixels and smaller lead to considerable attenuation of low frequencies as these components are not sampled in the procedure. Third, analysis of the 7 to 4 Å region reveals that the deviation from the exponential Wilson profile in this region is strongest at intermediate window sizes ranging from 20 to 35 Å. As these deviations are due to Debye effects originating from secondary structure this observation indicates the existence of an optimal window size for which these effects are maximized. Visual inspection of the resulting densities reveals overall similar appearance (*Figure 3—figure supplement 3B*; data shown representatively for V430-F449 helix), but window sizes ranging from 25 to 35 Å (20–30 pixels) appear to represent the expected features best. To assess whether the map series can be independently quantified for 'sharpness' we computed the map kurtosis for all maps. Kurtosis first increased, then decreased with increasing window size, peaking between 22–30 Å (*Figure 3—figure supplement 3C*). Maximal map kurtosis thus correlates with strongest deviation in the 7 to 4 Å region of the Guinier plot. As the extent of this deviation is related to both model perturbations and B-factors (*Figure 3—figure supplement 2D–I*), the effect can be expected to be maximized in maps that are optimally sharpened. Our results indicate that an optimal window size can be chosen that best reflects the effective distance of local B-factor correlations, thus maximizing map kurtosis and Debye effect contrast across the sharpened map.

## Integrating model-based sharpening with model building and refinement

Based on the demonstrated benefits of model-based sharpening, we propose to integrate local sharpening with atomic model refinement: First, an initial atomic model is generated and atomic coordinates and B-factors are refined using existing procedures (*Hoffmann et al., 2015*; *Murshudov, 2016*; *Wang et al., 2014*). In a second step, we use the refined coordinate model to generate a reference map used for local amplitude scaling of the cryo-EM density. The LocScale density contains enhanced local contrast and can be used as an improved target map for additional rounds of model building and refinement. The procedure can be iterated until convergence (*Figure 4A*).

One of the potential shortcomings of the proposed method is the principal requirement of an atomic model. The iterative character of integrating the local scaling procedure into building and refinement workflows, however, can be used to assist in extending an existing model to regions of the map where no reference model was available at first. We illustrate the use of this strategy on the TRPV1 channel where we started with an atomic model that only covered the transmembrane region and the extracellular density was not occupied (*Figure 4B*). As a density window of 25 Å was used for local scaling, the resulting LocScale density extended at least half the window size beyond the model until the density progressively disappeared. In this case, the level of sharpening for an adjacent density was approximately correct even in the absence of the complete model, and allowed the extension of the model. This model served as a reference for additional cycles of model building, refinement and sharpening until model completion.

## LocScale maps facilitate model building and reveal additional structural detail

We tested whether LocScale maps can aid model building and refinement using the EMDB model challenge cases TRPV1 channel, γ-secretase, β-galactosidase, ribosome EF-Tu complex, 20S proteasome and RNA PolIII. All systems pose unique challenges: Pol III display considerable variation in local resolution across the map ranging from good (<4 Å) to very low resolution (>8 Å; see above). The ribosome EF-Tu complex contains a large fraction of ribonucleic acid in addition to protein

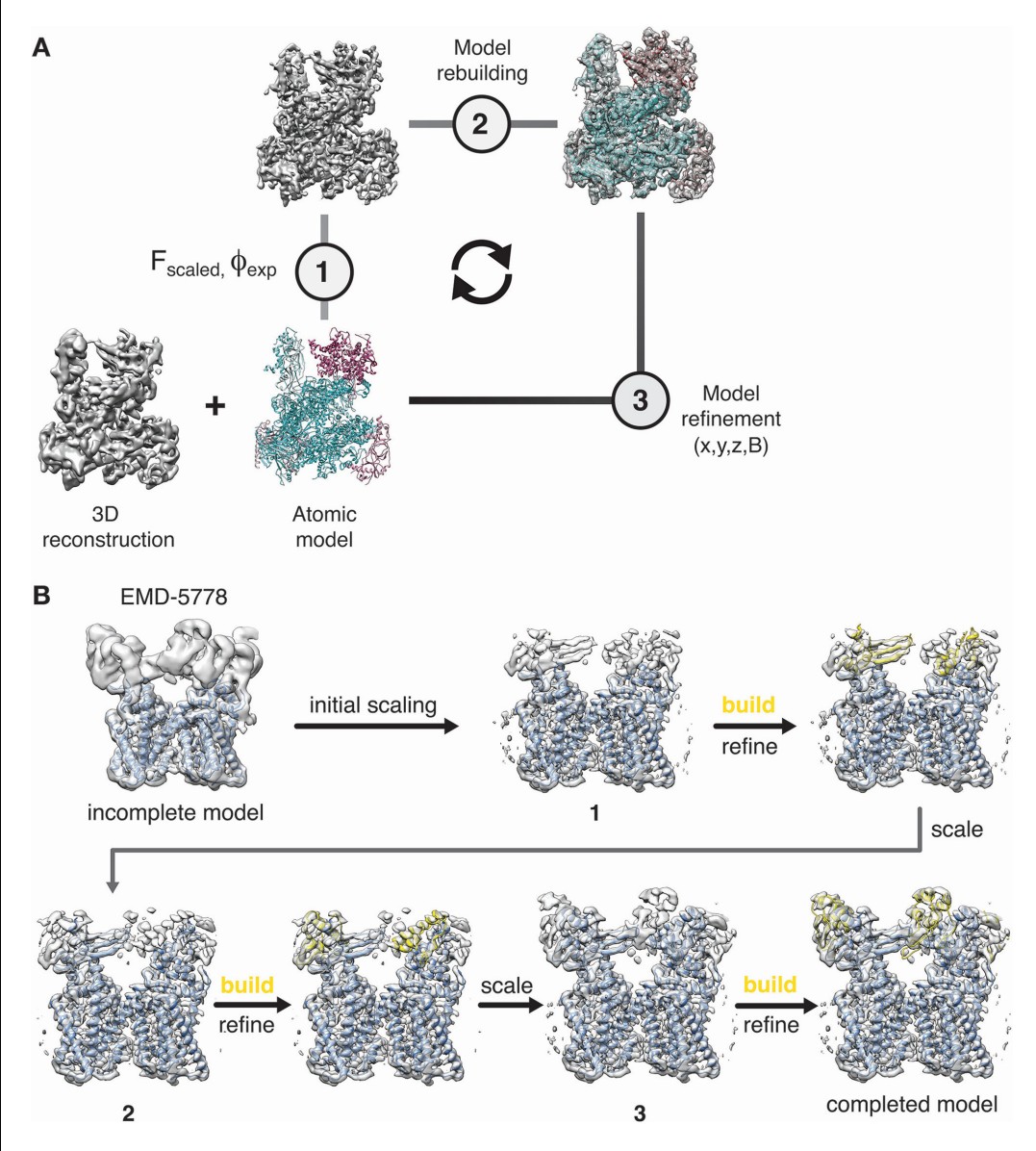

**Figure 4.** Cyclical workflow of iterative density sharpening using reference-based scaling and successive completion of atomic models. (**A**) An atomic model is used as a reference to improve contrast in the original cryo-EM reconstruction employing local amplitude scaling (1). The resulting LocScale map contains improved contrast and thus facilitates further model building (2) and coordinate refinement (3). Scaling, building and refinement steps can be iterated to produce successively improved density maps and atomic models. (**B**) Workflow for extending atomic model beyond the reference structure using the LocScale procedure. (Top left) EM density for TRPV1 (EMD-5778) superposed onto an incomplete atomic model covering only the transmembrane region. In three successive cycles of LocScale sharpening and extension of the initial coordinate set by model building into new density with improved contrast, the atomic model can be completed to interpret the entire experimental density from EMD-5778. An effective window size of 25 Å was used for the LocScale iterations.
DOI: https://doi.org/10.7554/eLife.27131.011

components. γ-secretase (*Bai et al., 2015*) contains large local resolution differences between the protein core and embedding amphipols. The TRPV1 channel (*Liao et al., 2013*) displays substantial resolution differences between the transmembrane and extracellular regions. β-galactosidase (*Bartesaghi et al., 2015*) is an example of a structure resolved at an exceptionally high resolution (2.2 Å). These examples allowed us to test the effectiveness of the procedure for a resolution range

from ~8 to 2 Å. To start with comparable reference models, we first generated uniformly filtered and sharpened maps and performed one round of atomic B-factor refinement using the deposited PDB coordinates as starting models. For further analysis, we computed LocScale maps using the resulting reference models as described above.

We first assessed the validity of our procedure by comparing the deposited map of β-galactosidase (EMD-2984) with the map obtained by the LocScale procedure. The density of EMD-2984 is of such exceptional quality that individual water molecules can be placed, and characteristic doughnut-shaped densities of aromatic rings are revealed in the best-resolved regions of the map (*Bartesaghi et al., 2015*). Visualizing these high-resolution features in the deposited map required sharpening levels that result in a density with high overall noise levels (*Figure 5A*). Qualitatively, the LocScale map appears less noisy when compared with the deposited map, which is supported by the smooth falloff of the radially averaged power spectrum profile (*Figure 5—figure supplement 1A*). At the same time, the LocScale map does preserve the reported high-resolution details such as ordered water molecules and characteristic density holes of aromatic rings (*Figure 5A–B*). While most of the structure shows an excellent fit between density and the deposited coordinate model (PDB ID 5a1a), we noted that one particular region (V728–H735; *Figure 5C* inset) displayed poor real-space correlation and the model in this region contains unusually high B-factors relative to other parts of the structure. Closer inspection of the deposited model and map revealed that density in this region is fragmented and precludes confident modelling of the backbone conformation (*Figure 5D*). By contrast, the LocScale density showed a clear backbone trace and reveals that the deposited coordinates in this region are incompatible with the density (*Figure 5E*). The LocScale map allowed us to rebuild this residue stretch to better satisfy the density (*Figure 5F*, *Figure 5—figure supplement 1B*). Notably, a comparable representation of density for this loop can also be obtained by blurring the original map with a B-factor of 50 Å$^2$ (*Figure 5F*, *Figure 5—figure supplement 1C*), but at the expense of detail in better resolved parts of the map. To verify our rebuilt conformation, we compared both PDB ID 5a1a and the rebuilt model with the 1.6 Å resolution crystal structure of *E.coli* β-galactosidase ([*Wheatley et al., 2015*], PDB ID 4ttg). The V728–H735 conformation built and refined using the LocScale map is in closer agreement with the high-resolution crystal structure than the deposited model (*Figure 5–figure supplement 1D,E*) and converges after two LocScale iterations alternating with model refinement (*Figure 5—figure supplement 1F*). In EMD-2984, the fragmented density and high noise level in this region appear to have misguided model building.

The β-galactosidase cryo-EM structure contains the inhibitor phenylethyl-β-D-thiogalactopyranoside (PETG) in the active site. When analyzing the ligand density in the active site binding pocket, we realized that the LocScale map showed additional density features suggestive of hydroxyl substituents at the hexa-pyranosyl ring that were in mismatch with the modelled ligand orientation in PDB 5a1a. Refinement against the LocScale map led to re-orientation of the ligand (*Figure 5—figure supplement 1H–J*) and several active site residues. Although the overall ligand shifts are small (0.7–1.0 Å, rmsd 1.468 Å over 20 atoms), the ligand conformation obtained with the LocScale map satisfies additional hydrogen bonds in the active site and results in tighter accommodation of the inhibitor in the binding pocket (*Figure 5—figure supplement 1K,L*).

We also generated LocScale maps for the ribosome EF-Tu complex, TRPV1, Pol III, γ-secretase and the 20S proteasome. Interestingly, when inspecting the γ-secretase LocScale map, we found additional density contrast at 9 out of 11 N-glycosylation sites, which would permit modeling glycan extensions beyond the deposited carbohydrate chains (*Figure 5G–I*). Since the density did not allow unambiguous assignment of chemical identity, we in these cases refrained from building additional carbohydrate residues. Notably, in addition to previously described sites, the LocScale map revealed interpretable density at four additional and previously unreported N-linked glycosylation sites that allowed building of the glycan base (*Figure 5J/K*).

One notable benefit of using locally scaled maps is that all parts of the map can be simultaneously visualized at a single threshold level. We illustrate this property for the ribosome EF-Tu complex (*Fischer et al., 2015*) (*Figure 6*). In particular, RNA nucleotides, ribosyl moieties and phosphate backbone often require substantially different density thresholds for adequate visualization, which can pose challenges to model building. While in all cases, suitable threshold levels to represent the respective density can be found locally within globally sharpened maps, visualization of weak detail comes at the expense of interpretability of adjacent parts of the structure. LocScale

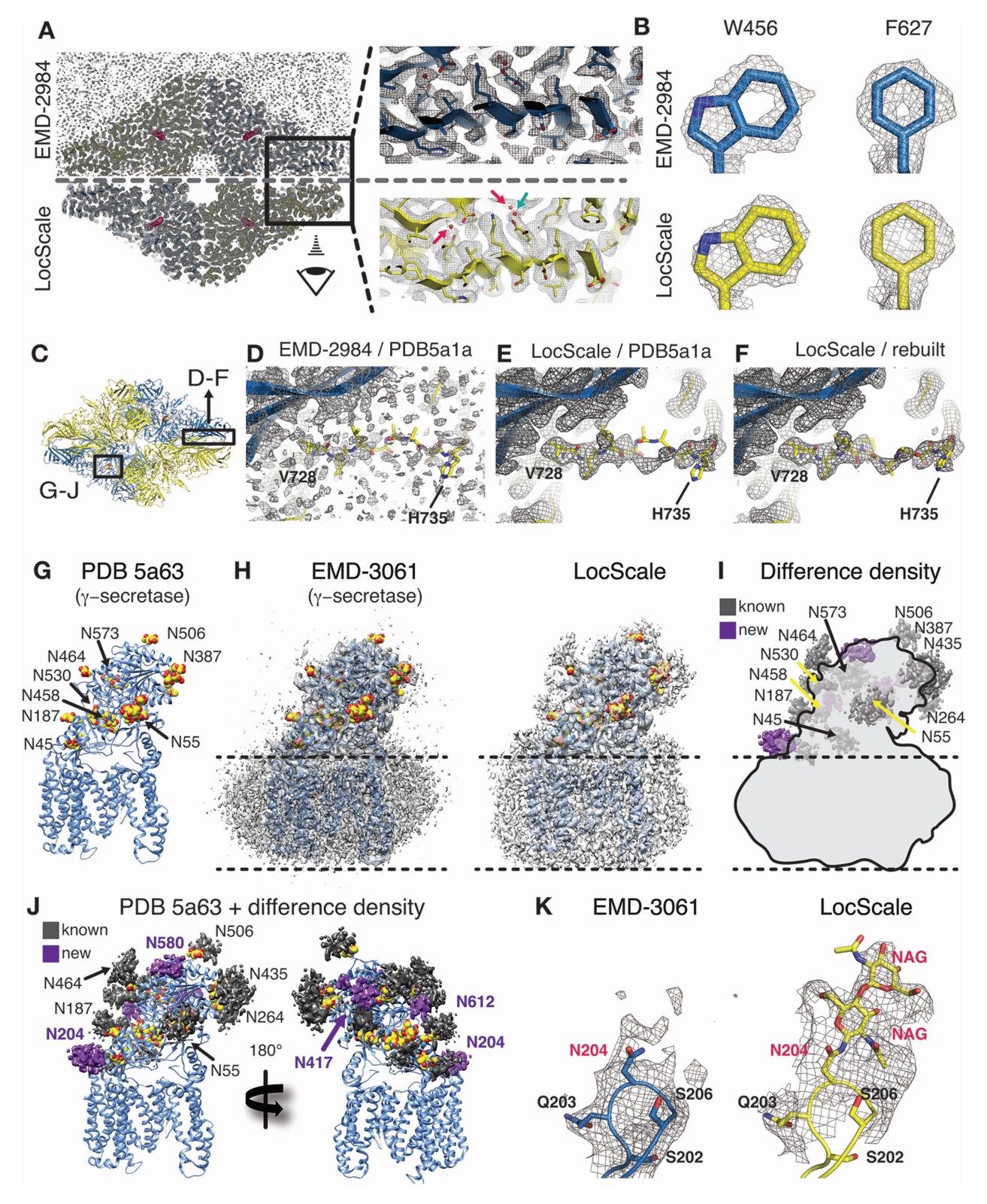

**Figure 5.** Application of the LocScale procedure to the 2.2 Å resolution cryo-EM structure of β-galactosidase and 3.4 Å structure of γ-secretase. (**A**) Overall comparison of uniformly sharpened (EMD-2984) (top) and LocScale map (bottom) for the β-galactosidase structure. EMD-2984 displays visible noise at the periphery of the molecule and its surrounding solvent, whereas the LocScale map contains reduced noise levels in corresponding regions. Zoomed-in regions in the protein core showing modeled water molecules for which density is retained in the LocScale map (arrows). (**B**) High-resolution
*Figure 5 continued on next page*

*Figure 5 continued*

details such as doughnut-shaped densities for aromatic side chains W456 and F627 in the EMD-2984 map (top) are consistently retained in the LocScale map (bottom). Maps are shown at equivalent isosurface levels displayed at a 1.6 Å carve radius. (**C**) Cartoon representation of the β-galactosidase tetramer (PDB ID 5a1a) colored by symmetry-related subunits (yellow and blue). (**D–F**) Density around the peripheral loop region comprising residues V728– H735 (see (**C**) for overall location) fragmented in the deposited map, hampering unambiguous tracing of the backbone conformation. Deposited conformation in PDB ID 5a1a superimposed on the EMD-2984 map (**D**) and LocScale map (**E**). Note reduced noise levels and continuous density providing an unambiguous backbone trace in the LocScale map. The LocScale density is not compatible with the modelled conformation of PDB ID 5a1a, but supports an alternative conformation of V728–H735 shown superimposed on the LocScale map (**F**). (**G**) Atomic model of γ-secretase (PDB ID 5a63) in cartoon representation. Modeled glycan residues in nicastrin are shown as spheres and N-linked glycosylation sites are indicated. (**H**) Comparison of EMD-3061 (left) and LocScale (right) maps contoured at 3σ with ribbon model as in (**G**). (**I**) Difference density of (H left and right) superposed on light gray silhouette representation of EMD-3061. Regions with difference density beyond 4σ located at known glycosylation sites are colored in dark gray and newly identified glycosylation sites are shown in purple. (**J**) Difference density from (**I**) mapped onto the atomic model of γ-secretase. Purple difference density locates to putative additional glycosylation sites. (**K**) Density around N204 of nicastrin for EMD-3061 (left) and LocScale map (right) reveals additional density for two N-acetylglucoamine (NAG) residues in the LocScale map.

DOI: https://doi.org/10.7554/eLife.27131.012

The following figure supplement is available for figure 5:

**Figure supplement 1.** Comparison of β-galactosidase radial amplitude profiles and iterative refinement of coordinates from the V728–H735 loop region.

DOI: https://doi.org/10.7554/eLife.27131.013

maps allow visualization of amino acid and nucleotide densities at a single threshold simultaneously (*Figure 6G,H*). The most significant differences between overall and locally sharpened maps are observed in maps displaying large resolution variations, such as RNA Pol III (*Figure 6—figure supplement 1*). Similar observations can be made for peripheral densities in the 20S proteasome structure (*Campbell et al., 2015*) that are substantially weaker than the better resolved core structure (*Figure 6—figure supplement 2*). For both TRPV1 and Pol III, the respective LocScale maps provided enhanced contrast in regions of previously fragmented and weak density. The LocScale maps allowed the rebuilding of loop regions and side chains in the TRPV1 structure (*Figure 7A–C*) and confident placement of predicted secondary structure elements in previously less interpretable parts in the C82 subunit of Pol III (*Figure 7D–F*). Together, these examples showcase the potential of locally enhanced contrast in amplifying weak but relevant map features and the ability to simultaneously visualize all parts of a map at comparable contrast to facilitate map interpretation and model building.

## LocScale maps are improved map targets for model refinement

In order to assess quantitatively whether model-based local sharpening positively affects atomic model refinement, we performed coordinate refinement using either the EMDB-deposited or the LocScale map as minimization targets. Initially, we randomly perturbed the coordinates of the PDB-deposited coordinates of all four structures (β-galactosidase: PDB 5a1a; TRPV1: PDB 3j5p; Pol III: PDB 5fja; γ-secretase: PDB 5a63) and subjected the resulting models to iterative real-space refinement against the EMDB-deposited map and the corresponding LocScale map (*Adams et al., 2010*; *Hoffmann et al., 2015*). In every case, the overall real-space correlation improved by 3–6% for models refined against the LocScale map when measured against the deposited EMDB entry (*Table 1*), and FSC curves computed between model maps and the original reconstructions show small but notable improvements of the LocScale model fit (*Figure 7—figure supplement 1*). It should be noted that comparisons of real-space correlation and model-map FSC measures are only meaningful when they are calculated against a common reference volume that has not undergone model-based scaling. FSC improvements are largest for TRPV1 and Pol III, for which also the resolution differences across the map are largest. In addition, we find improved EMRinger and Molprobity scores (*Barad et al., 2015*; *Chen et al., 2010*) for all models refined with LocScale maps, indicating that better density fit is not achieved at the expense of model distortions.

To better discriminate the individual effects of low-pass filtration and local contrast enhancement on real-space refinement targets, we also examined uniformly sharpened, local resolution-filtered (LocRes) maps (see Materials and methods) as targets for atomic model refinement. We observed that LocRes maps partially compensate for artifacts arising from uniform sharpening in 3D

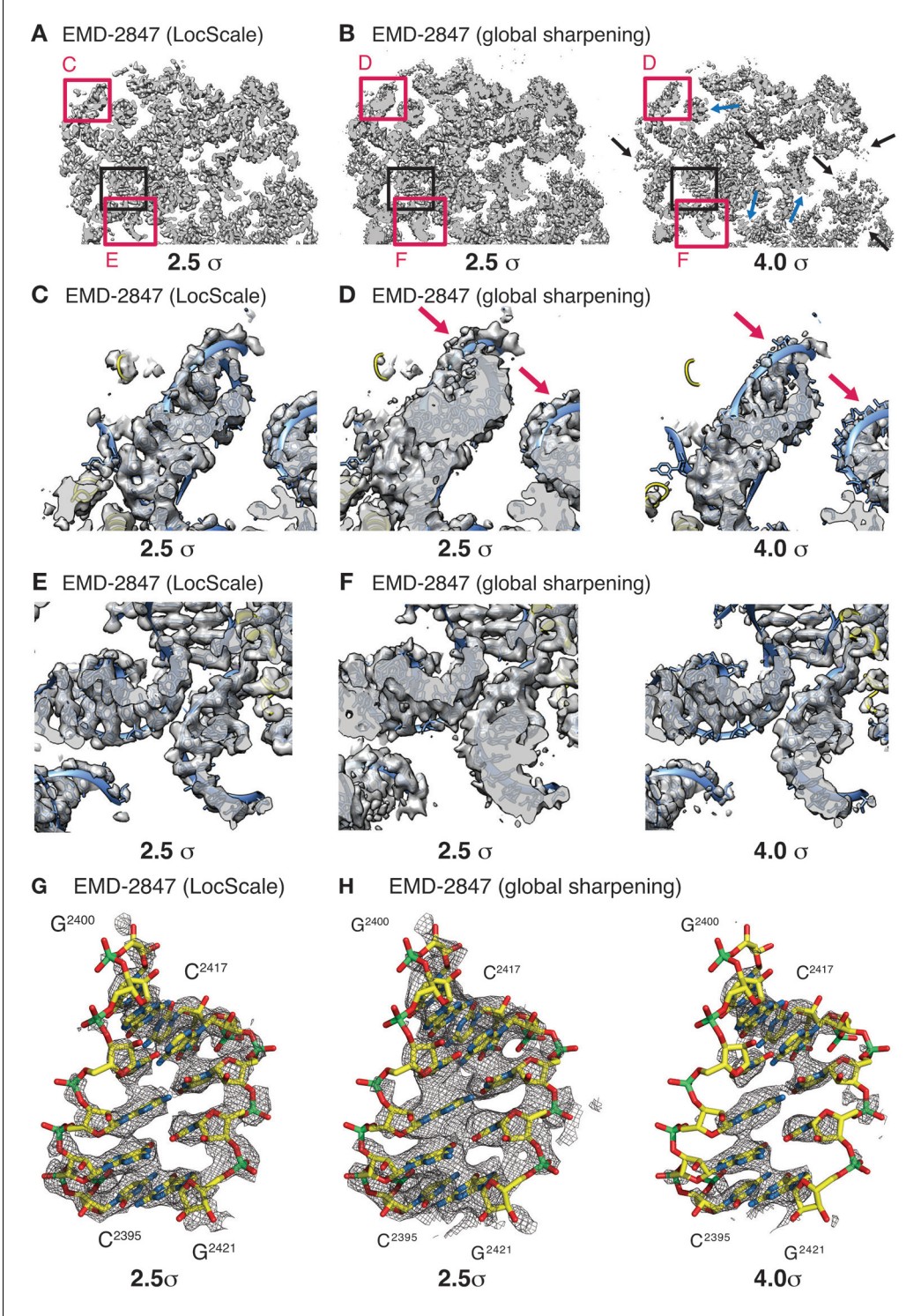

**Figure 6.** Effect of local sharpening on RNA density in the ribosome EF-Tu complex. (**A**) Cross-section through cryo-EM density of the *E.coli* ribosome EF-Tu complex (EMD-2847) after local sharpening with LocScale contoured at 2.5 σ. Red frames correspond to density enlarged in panels C and E whereas black frame corresponds to density in G. All parts of the LocScale map are interpretable at a single density threshold level. (**B**) Cross-section as in (**A**) after sharpening with a global B-factor contoured at 2.5 σ (left) and 4.0 σ (right). Not all parts of the maps are equally interpretable at 2.5 σ. Strong density sections reveal increased detail at higher threshold levels (blue arrows). In turn, weaker parts of the map are no longer visible at these threshold levels (black arrows). (**C**) Close-up of LocScale density displaying a peripheral ribosomal RNA fragment as indicated in (**A**). RNA density including

*Figure 6 continued on next page*

*Figure 6 continued*

nucleotide bases and the phosphate backbone is visible (blue ribbon) and base stacking is partially recognizable along the RNA strand. In addition, protein density (yellow ribbon) is visible when contoured at 2.5 σ. (D) Same area as in (C), here displayed for the globally sharpened map contoured at 2.5 σ (left) and 4.0 σ (right). Visualization at increased density threshold is required to reveal additional detail on nucleotide base stacking, at the expense of density for phosphate backbone (arrows) and adjacent protein components. (E) Close-up of LocScale density displaying a well-resolved ribosomal RNA fragment as indicated in (A). Nucleotide base stacking is recognizable in all displayed RNA duplexes. (F) Same area as in (E) displayed for the globally sharpened map contoured at 2.5 σ (left) and 4.0 σ (right). (G) LocScale density: detailed view on a ribosomal RNA fragment (shown in stick representation; N (blue) O (red), C (yellow), P (green)) for EMD-2847. The density is contoured at 2.5 σ. (H) Globally sharpened density: same RNA fragment as in (G) for EMD-2847 contoured at 2.5 σ (left) and 4.0 σ. Simultaneously resolving nucleotide base stacking and phosphate backbone at a single threshold level is not possible. A carve radius of 2.5 Å was used for density representations in G and H.

DOI: https://doi.org/10.7554/eLife.27131.014

The following figure supplements are available for figure 6:

**Figure supplement 1.** Effect of local sharpening on threshold level and map interpretability of maps with resolution variation illustrated for RNA Pol III.

DOI: https://doi.org/10.7554/eLife.27131.015

**Figure supplement 2.** Effect of local sharpening on peripheral density in the 20 s proteasome.

DOI: https://doi.org/10.7554/eLife.27131.016

reconstructions with significant resolution variation, whereas local contrast optimization in LocScale maps consistently provides additional detail (*Figure 7—figure supplement 2*). Comparison of global refinement statistics using LocRes and LocScale maps as targets support these qualitative observations (*Table 2*). Together, the results from four examples show that local contrast optimization by LocScale sharpening results in atomic models with improved overall density fit and good stereochemistry.

## Discussion

We have introduced a procedure to improve the interpretability of cryo-EM density maps by model-based local sharpening. The LocScale method uses prior information from an atomic reference model that itself improves progressively during rounds of alternating coordinate refinement and LocScale sharpening. Our method is generally applicable to any cryo-EM structure whose density can be interpreted in terms of an atomic model and thus only requires the input of density map and superimposed atomic coordinates without further specification of tuning parameters. Current contrast enhancing approaches often inadvertently result in either omission of structural detail or amplification of background noise, resulting in map artifacts that often cannot be easily distinguished from real atomic features. By applying amplitude scaling locally, LocScale maps represent high-resolution signal more accurately throughout the map (*Figures 2–7*). The procedure is reminiscent of an optimized low-pass filter that adaptively scales and filters the map amplitudes to match the local resolution and expected mass density distribution of the underlying molecular structure. Instead of reference-based scaling as implemented in the LocScale procedure, local sharpening can also be performed after estimating the amplitude falloff within each density window using Guinier fitting. While this approach, along with UNSHARP masking, remains a viable alternative for cases where no model is available, or is largely incomplete, we find that reference-based scaling resulted in improved detail compared with local Guinier scaling. Strictly speaking, exponential sharpening is based on an approximation that assumes scattering to arise from randomly distributed free atoms (*Wilson, 1942*). One advantage of LocScale amplitude scaling over other sharpening methods is that it explicitly accounts for the characteristic local profile of the amplitude falloff that is due to specific structural properties of secondary structure motifs (*Morris et al., 2004*) and can differ markedly from a simple exponential decay (*Figure 2F*, *Figure 2—figure supplement 1*).

We find that the magnitude of these Debye effects positively correlates with kurtosis, where the latter represents a quantification of the apparent 'sharpness' of the density. Confirmed by visual inspection, we found that map kurtosis represents a map quality indicator for cryo-EM densities similar as proposed previously for X-ray crystallographic maps (*Afonine et al., 2015*). Using six

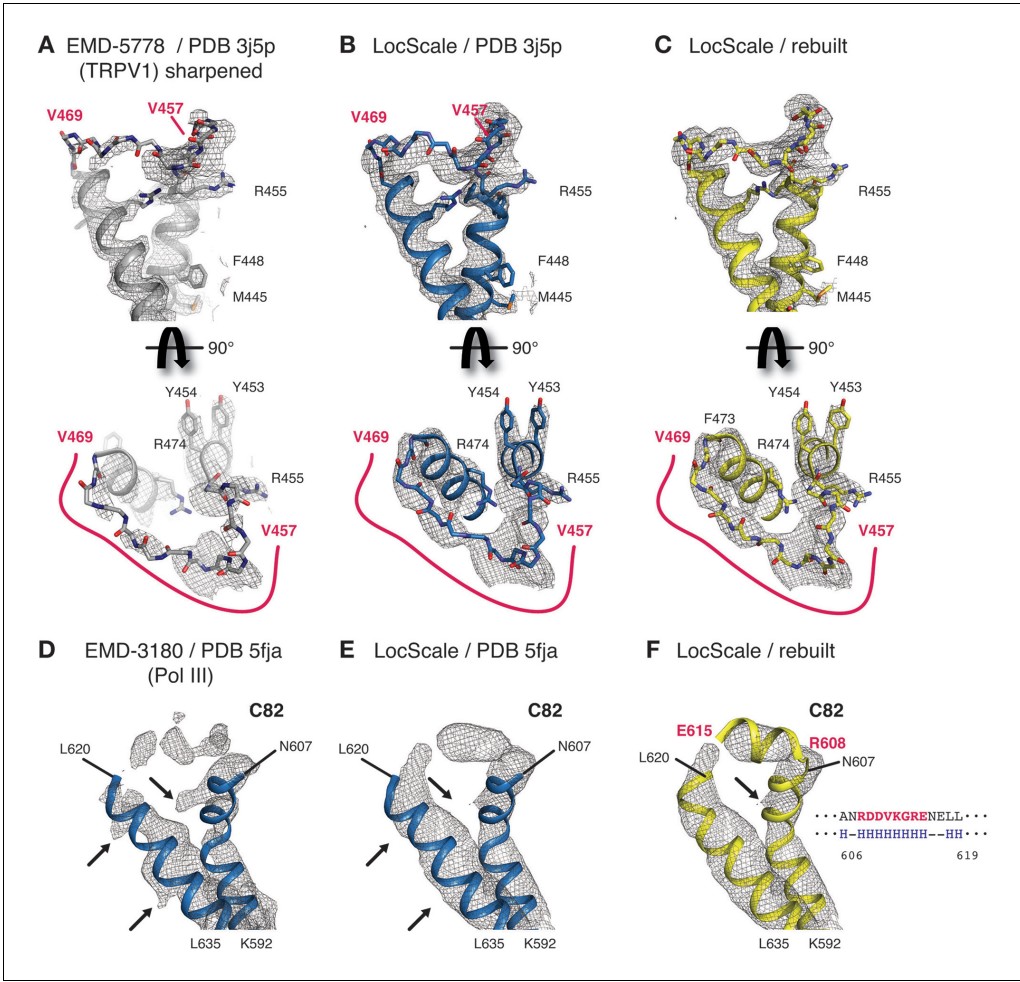

**Figure 7.** Details of LocScale maps for TRPV1 and RNA Pol III. (**A**) Close-up density for TRPV1 channel (EMD-5778) superposed on the deposited coordinate model (PDB ID 3j5p). The map was additionally sharpened with a B-factor of −100 Å² to make densities for aromatic and other large side chains more clearly visible. Transmembrane views (top) and the corresponding perpendicular view (bottom) of the V457–V469 loop region are displayed, showing poorly resolved density for V457–V469 residues (main chain shown in stick representation). Selected residues with large side chains are also shown in stick representation. (**B**) Same view as (**A**) with PDB ID 3j5p superposed onto the LocScale density. Density is shown at a threshold level matching visible side-chain densities in (**A**). Continuous density is observed for residues V457–V469, allowing less ambiguous building of the main chain trace. (**C**) LocScale density for the same regions shown in (**A**) and (**B**) superposed on the atomic model obtained after adjustment and refinement against the LocScale map. (**D**) Close-up density for Pol III subunit C82 (EMD-3180) superposed onto the deposited atomic model (PDB 5fja) shown in cartoon representation. Density for residues R608–L619 is fragmented and precludes model building. Note the pronounced density artifacts (arrows) incompatible with side chain densities. (**E**) Same view as (**D**) with PDB 5fja superposed onto the LocScale density. The density is suggestive of a helical conformation for residues R608–E615, confirming secondary structure predictions (right). Density artifacts from (**D**) are reduced in the LocScale map. (**F**) C82 model rebuilt and refined using the LocScale map. The predicted helix formed by residues R608-E615 fits the helical density as suggested by the LocScale map.

DOI: https://doi.org/10.7554/eLife.27131.017

The following figure supplements are available for figure 7:

**Figure supplement 1.** Model validation.
DOI: https://doi.org/10.7554/eLife.27131.018

**Figure supplement 2.** LocScale maps recover contrast better when compared with local resolution-filtered (LocRes) maps.
DOI: https://doi.org/10.7554/eLife.27131.019

**Table 1.** Model refinement statistics

| | β-galactosidase (*Bartesaghi et al., 2015*) | | TRPV1 Channel (*Liao et al., 2013*) | | RNA polymerase III (*Hoffmann et al., 2015*) | | γ-secretase (*Bai et al., 2015*) | |
|---|---|---|---|---|---|---|---|---|
| | EMD-2984 | LocScale | EMD-5778 | LocScale | EMD-3180 | LocScale | EMD-3061 | LocScale |
| Starting model PDB ID | 5a1a | | 3j5p | | 5fja | | 5a63 | |
| Resolution (Å) LocRes range (Å) | **2.2** (1.9–3.6) | | **3.4** (2.5–4.8) | | **4.7** (3.4–8.1) | | **3.4** (3.0–6.8) | |
| Refinement (Å) | 186.0–1.9 | | 311.3–2.5 | | 320.9–3.4 | | 252.0–3.0 | |
| Overall RSCC* | 0.70 | 0.75 | 0.62 (0.75)§ | 0.67 (0.81)§ | 0.70 | 0.76 | 0.71 | 0.77 |
| EMRinger score† | 4.03 | 4.90 | 0.24 | 0.77 | | | 2.60 | 2.87 |
| MOLPROBITY score | 2.04 | 1.86 | 2.69 (1.44)§ | 1.98 (1.35) | 2.68 | 2.24 | 2.01 | 1.81 |
| Clash score‡ (all atoms) | 6.63 | 3.14 | 6.88 (2.42)§ | 5.29 (2.09)§ | 16.44 | 11.04 | 6.44 | 5.89 |
| Rotamer outliers (%) | 3.32 | 4.92 | 16.38 (0.00)§ | 2.23 (0.00)§ | 2.53 | 1.48 | 1.53 | 0.86 |
| Ramachandran statistics *Favored (%)* *Disallowed (%)* | 95.98 0.00 | 96.57 0.00 | 94.3 (93.6)§ 0.35 (0.64)§ | 93.4 (94.6)§ 0.17 (0.64)§ | 83.2 1.09 | 85.4 0.99 | 92.86 0.33 | 91.74 0.33 |
| RMS bonds (Å) | 0.008 | 0.01 | 0.008 | 0.006 | 0.008 | 0.01 | 0.011 | 0.012 |
| RMS angles (°) | 1.41 | 1.50 | 1.71 | 1.44 | 1.67 | 1.32 | 1.41 | 1.53 |

*Overall real-space correlation computed at the average map resolution using a soft mask around atoms. The EMDB deposition was used as the reference map in all cases.

†EMRinger score calculated only for parts of structure with local resolution estimate better than 4.5 Å.

‡Clash score denotes number of van der Waals overlap per 100 atoms.

§EMRinger scores in brackets are reported for the transmembrane region.

DOI: https://doi.org/10.7554/eLife.27131.020

**Table 2.** Model refinement statistics for local resolution-filtered maps.

| | β-galactosidase | TRPV1 channel | RNA polymerase III | γ-secretase |
|---|---|---|---|---|
| | LocRes | LocRes | LocRes | LocRes |
| Starting model PDB ID | 5a1a | 3j5p | 5fja | 5a63 |
| Resolution (Å) LocRes range (Å) | **2.2** (1.9–3.6) | **3.4** (2.5–4.8) | **4.7** (3.4–8.1) | **3.4** (3.0–6.8) |
| Refinement (Å) | 186.0–1.9 | 311.3–2.5 | 320.9–3.4 | 252.0–3.0 |
| Overall RSCC* | 0.72 | 0.64 (0.78)§ | 0.72 | 0.73 |
| EMRinger score† | 4.15 | 0.46 | 0.89† | 2.69 |
| MOLPROBITY score | 2.00 | 2.43 (1.40)§ | 2.62 | 1.95 |
| Clash score‡ (all atoms) | 4.45 | 6.45 (2.34)§ | 13.31 | 6.18 |
| Rotamer outliers (%) | 3.83 | 4.63 (0.00)§ | 1.95 | 1.00 |
| Ramachandran statistics *Favored (%)* *Disallowed (%)* | 96.24 0.00 | 95.1 (94.9) 0.31 (0.64)§ | 86.4 1.00 | 93.01 0.48 |
| RMS bonds (Å) | 0.008 | 0.008 | 0.01 | 0.01 |
| RMS angles (°) | 1.48 | 1.43 | 1.42 | 1.50 |

*Overall real-space correlation computed at the average map resolution using a soft mask around atoms. The EMDB deposition was used as the reference map in all cases.

†EMRinger score calculated only for parts of structure with local resolution estimate better than 4.5 Å.

‡Clash score denotes number of van der Waals overlaps per 100 atoms.

§Scores in brackets are reported for the transmembrane region.

DOI: https://doi.org/10.7554/eLife.27131.021

examples, we showed that iterative LocScale contrast restoration consistently facilitates interpretability of density maps and accuracy of atomic models over a range of resolutions between 2 and 8 Å. Notably, we also find LocScale maps to facilitate visualization of detail in nucleic acid structures. It is commonly observed in high-resolution cryo-EM structures containing ribonucleotide chains that the density associated with phosphate groups is weaker than that associated with the corresponding nucleotide bases. The effect can be partially compensated using reference scaling. The result is a map in which phosphate backbone and nucleotides can be visualized simultaneously with comparable detail at equivalent thresholds, thus facilitating model building.

While we have only tested the utility of such maps for real-space refinement, we expect LocScale maps to equally benefit phase-restrained refinement in reciprocal space as implemented in CNS (*Brunger et al., 1998*) or REFMAC (*Murshudov et al., 1997*). Although LocScale presents a form of model-based density improvement, it is important to distinguish our procedure from what is commonly understood by density modification in X-ray crystallographic refinement. Crystallographic density modification improves map quality through obtaining new phase estimates by modification of the electron density to match expected properties such as flatness or disorder of the solvent region (*Cowtan, 2010*; *Wang, 1985*; *Zhang et al., 1997*). In the LocScale approach put forward here, the amplitude component of density is modified such that it conforms to prior expectations from a reference model, whereas the phase component is retrieved directly from the experiment and remains unchanged by the procedure. As the method is restricted to scaling of amplitudes against a radial average, we argue that reference coordinate bias from the atomic model is not present. We show that for cases with incomplete models or moderate conformational or register errors in the model, the information stored in the radial amplitude profile still provides a good initial estimate of the mass density distribution and is sufficient to restore low contrast map features more clearly when compared with overall sharpening procedures (*Figure 3E–F*, *Figure 3—figure supplement 1A–E*). We find that either uniformly sharpened or globally scaled maps, optionally combined with filtering based on local resolution estimates, are appropriate starting targets for initial model building and refinement of atomic coordinates and B-factors. Once estimates of atomic B-factors have been obtained, LocScale maps provide locally optimized contrast and facilitate further rounds of model building and refinement. While it has been shown that map sharpening positively affects model refinement (*Afonine et al., 2015*; *Joseph et al., 2016*), a recent report suggested that overall benefits are limited (R. Y.-R. *Wang et al., 2016*). In our test cases with different ranges of resolution, we do confirm previously described benefits of sharpening also for the LocScale maps, likely because locally optimized sharpening levels result in improvements of density fit and geometry over all map areas.

Regardless of resolution, cryo-EM maps degrade in quality towards the particle periphery due to limitations of current alignment and 3D reconstruction algorithms. Furthermore, true structural flexibility is expected to be more prominent in the particle periphery as vitrification captures the solution-state conformational landscape at the respective experimental condition. While part of this conformational heterogeneity can be dealt with in unsupervised 2D and 3D image classification procedures (*Jonić, 2017*; *Lyumkis et al., 2013*; *Scheres, 2012*), remaining flexibility degrades resolution and results in localized blurring of the density map. The need for approaches that address resolution variation has been recognized and local resolution filtering has been proposed (*Cardone et al., 2013*; *Fernandez-Leiro and Scheres, 2017*; *Moriya et al., 2017*). The LocScale procedure takes the concept of local filtering further by optimizing contrast of the cryo-EM density through local amplitude scaling against an atomic reference model. In the six presented test cases, contrast improvements were particularly apparent in regions of increased structural flexibility such as peripheral subunits or flexible loop regions, as well as partial occupancy sites of ligands, active site residues and carbohydrate chains at N-linked glycosylation sites. It is not uncommon that such parts of protein complexes contain important information about the inhibition mechanism and interaction interfaces. Atomic model refinement followed by a local amplitude scaling step results in a single map that displays structural information optimally for all map areas simultaneously, including those parts with large resolution variation.

Due to the high number of voxels present in a 3D EM density map, our procedure requires a large number of parameters and poses a risk of overfitting. For every voxel, a map B-factor and a FOM combined in a sharpening profile need to be determined by scaling the experimental to the reference radial amplitude profiles. Our implementation has two built-in safety features that prevent

free fitting of the two parameters for every voxel. First, the radial amplitude profile of the reference is estimated from a restrained model refinement procedure that can be assessed for overfitting. These procedures use established protocols for restraining stereochemistry and atomic B-factors that have been routinely employed in crystallographic refinement. Second, the chosen window size ensures that radial averaging over extended density regions, leading to implicit correlations between scale factors over a certain distance. Consequently, the two fitted parameters are kept locally restrained, thus reducing the risk for overfitting of local sharpening profiles.

For best performance, LocScale sharpening requires good estimates of atomic B-factors associated with the reference coordinates. We show, however, that local scaling is sufficiently robust to result in improved maps even with reference structures for which atomic B-factors have not yet been refined (*Figure 3—figure supplement 2*). Refinement of B-factors at low resolution is challenging and includes the risk of overfitting as the number of experimental observables is typically much lower than the number of parameters to be fitted (*DeLaBarre and Brunger, 2006*). Considering that atomic B-factors represent the square amplitude of displacement from the refined coordinates they correlate with resolution. We thus propose to use the agreement of B-factor and local resolution estimated on a per-residue basis as a low-level assessment tool for the robustness of the atomic B-factor refinement (*Figure 3—figure supplement 1G,H*). Finding better ways to refine and validate B-factors at low resolution in real and reciprocal space remains an important area of investigation with potential for further improvements in X-ray crystallographic as well as in cryo-EM model refinement (*Brunger et al., 2009*). In the test cases investigated here, we found atomic B-factor refinement to work acceptably up to a resolution of ~8 Å. Therefore, we expect the LocScale sharpening to work best for near-atomic resolution cryo-EM reconstructions that are appropriate for model building and allow reliable coordinate refinement including B-factors. Interestingly, inspection of $FSC_{ref}$ curves often displays notable disagreement between $FSC_{ref}$ and half-map FSCs at low resolution despite good real-space density correlations between the model and density map, indicating that additional efforts are required to accurately model molecular structure factors including solvent contributions (*Clarage et al., 1992*; *Jiang and Brunger, 1994*); and the need for inclusion of more sophisticated B-factor models to reflect concerted molecular motion of atom groups in cryo-EM model refinement strategies (*Painter and Merritt, 2006*).

For the future of cryo-EM, it is tempting to speculate that atomic models will become more tightly integrated into the structure determination process. In principle, up-stream parts of the structure determination process such as the orientation search of the particle images for 3D reconstruction may also benefit from efforts that aim at iteratively improving both density map and atomic model, analogous to X-ray crystallography procedures that refine the atomic model against diffraction data. We here put forward an alternative procedure for iterative cryo-EM density sharpening that makes use of local amplitude scaling derived from a refined atomic reference model. In our examples, the LocScale procedure results in more faithful density representations and at the same time overcomes previous shortcomings in dealing with maps displaying considerable resolution variation. The approach is expected to benefit a wide range of cryo-EM structures and will facilitate atomic model interpretation of the rapidly growing number of high-resolution cryo-EM density maps.

## Materials and methods

### Implementation

All computational methods have been written in Python (www.python.org) and make use of libraries and functions from SPARX/EMAN2 (*Hohn et al., 2007*) and CCTBX (*Adams et al., 2010*). Numerical operations are performed using the Numpy package and Scipy libraries (www.scipy.org). The program LOCSCALE and associated scripts including instructions are available for download at https://git.embl.de/jakobi/LocScale (*Jakobi and Sachse, 2017*). A copy is archived at https://github.com/elifesciences-publications/LocScale.

### Model-based amplitude scaling (LocScale maps)

The test image used in *Figure 2A–E* is in the public domain and the amplitude manipulations were implemented in Numpy and Scipy (www.numpy.org). For local amplitude scaling of cryo-EM

densities, model maps were obtained from atomic coordinates by computing B-factor weighted structure factors using electron atomic form (*Colliex et al., 2006*) and their conversion into density by inverse Fourier transform using a CCTBX-based routine (pdb2map.py). LOCSCALE accepts unfiltered and unsharpened volumes, that is untreated 3D reconstructions, for scaling. Experimental and model maps were sampled in a rolling window of (n x n x n) voxels. For each grid step, the unfiltered experimental density map contained within the rolling window was sharpened by a resolution-dependent amplitude scaling factor using

$$F_{corr}(s) = k(s) F_{obs}(s), \tag{1}$$

where $F_{obs}(s)$ is the Fourier amplitude of the original density map at spatial frequency $s$, and

$$k(s) = \sqrt{\frac{\sum_{s \pm \Delta s/2} |F_{model}|^2}{\sum_{s \pm \Delta s/2} |F_{obs}|^2}} \tag{2}$$

is the scale factor at frequency $s$ derived from the ratio of radially averaged power spectra integrated within a Fourier shell of thickness $\Delta s$. $F_{model}(s)$ and $F_{obs}(s)$ indicate Fourier amplitudes from model and experimental map, respectively. The map value of the central voxel contained in the scaled volume element is then assigned to the corresponding voxel of the new map. Repeating the procedure over each voxel position yields a map representation with optimized contrast.

The program accepts an unfiltered 3D reconstruction and a superposed atomic model as input. Alternatively, a precomputed model map can also be provided alongside the unfiltered 3D reconstruction. A soft-edged mask can be supplied to restrict the computations to the voxels contained within the mask. The program requires *dmin* and *window_size* as additional input parameters. The parameter *dmin* refers to the resolution up to which the reference map should be simulated and scaling should be performed (typically Nyquist frequency) and *window_size* denotes the pixel length of the density cube contained in the rolling window. As a typical *window_size* parameter, we used seven times the effective resolution (FSC 0.143 cutoff) in pixels in an effort to balance small window size to preserve locality of the approach against large window size to have sufficient number of Fourier pixels to appropriately sample the fine structure of the amplitude falloff.

## Local B-factor sharpening based on Guinier fitting

Map sharpening by local B-factor estimation from Guinier fitting was implemented in LOCSCALE using the same rolling density window framework described above. In this case two unfiltered half maps are provided as input. For each grid step the FSC between corresponding Hann-tapered density windows from either of respective half maps is computed. The density cubes are zero-padded before computing the Fourier transform to more accurately determine the local resolution estimate defined by a FSC threshold or $3\sigma$ criterion. Both density windows are then added and the logarithm of the spherically averaged amplitude against the squared spatial frequency is used to determine the local B-factor by least squares fitting. For fitting, the frequency range from [0.1 Å$^{-1}$ – d$_{min}$] is taken into account, where d$_{min}$ is the spatial frequency determined by a local resolution criterion estimated from the FSC curve. The density is subsequently sharpened by an amplitude correction term that is FOM-weighted according to the locally estimated signal-to-noise ratio using

$$F_{corr} = C_{ref} \, F_{obs} \, e^{-\left(B_{local}/4d^2\right)} \tag{3}$$

where B$_{local}$ is the negative of the locally estimated B-factor and

$$C_{ref} = \sqrt{\frac{2FSC_{local}}{1 + FSC_{local}}} \tag{4}$$

The map value of the central voxel contained in the scaled volume element is then assigned to the corresponding voxel of the new map.

## Unsharp masking

Unsharp masking was applied as implemented in CCTBX and described previously in (*Afonine et al., 2015*).

## Atomic model refinement

Unfiltered reconstructions for TRPV1 (EMD-5778), γ-secretase (EMD-3061) and β-galactosidase (EMD-2984) were obtained from the EMDB model challenge website (http://challenges.emdata-bank.org). Initial map targets for coordinate refinement were generated by applying uniform filtering to the original EMDB entries. The deposited map for TRPV1 (EMD-5778) was globally sharpened using an additional B-factor of $-100$ Å$^2$. Coordinate refinement was performed using real-space refinement against the initial map as implemented in cctbx/PHENIX (*Adams et al., 2010*) with additional restraints on secondary structure. Residue-grouped atomic B-factors were refined in reciprocal space. For TRPV1 and β-galactosidase, non-crystallography symmetry (NCS) restraints were employed to account for rotational and dihedral symmetry. Reference restraints for PETG were obtained with *phenix.elbow* from the crystal structure of 2-phenylethyl 1-thio-β-D-galactopyranoside (*Brito et al., 2011*). Building of glycans used glycan modeling tools available in Coot. Local resolution for all maps was estimated by local FSC computations using an in-house Python program *locres. py*. Local resolution mapped to the coordinate models using the *measure_mapValues* function in UCSF Chimera (*Pettersen et al., 2004*), and the residue-averaged resolution was used to assign local resolution-scaled reference restraints. Model rebuilding was performed in Coot (*Emsley and Cowtan, 2004*). Half-set reconstructions for cross validation were obtained from the EMDB model challenge. Model maps were generated by inverse Fourier transform using B-factor-weighted electron form factors (*Colliex et al., 2006*). The FSC$_{ref}$ between model map and half-set 3D reconstructions was used to assess over-fitting. To this end, coordinates were first refined against the full map. Coordinates were then randomly displaced by a maximum of 0.5 Å and subjected to three cycles of real-space refinement against one of the half maps (work map) using the same protocol outlined above. The other half map (test map) was used as the test map for cross-validation. Following refinement, we computed the FSC of the refined model against the work map (FSC$_{work}$) and the cross-validated FSC between refined model and the test map (FSC$_{test}$). All FSCs were computed using a structure mask obtained from the respective EMDB entry that was low-passed filtered to 60 Å.

To compare model refinement against deposited EMDB entries and LocScale maps, PDB-deposited coordinate models associated with the respective EMDB entry were randomly perturbed by applying atom shifts of up to 0.4 Å to serve as starting models. Five iterations of local and global real-space coordinate refinement against the respective EMDB or LocScale map were each followed by refinement of atomic B-factors in reciprocal space. As above, secondary structure restraints, resolution-dependent weights and NCS restraints were employed. EMRinger scores were computed using *phenix.em_ringer* (*Barad et al., 2015*); all other validation scores were obtained using MOL-PROBITY (*Chen et al., 2010*). All cross-validation FSC and real-space correlations were computed against the original reconstruction and the deposited EMDB map, respectively.

## Acknowledgements

AJJ acknowledges financial support by an EMBL Interdisciplinary Postdoc (EIPOD) fellowship under Marie Curie Actions (PCOFUND-GA-2008–229597), a Marie-Sklodowska-Curie IEF fellowship (PIEF-GA-2012–331285), the Deutsche Forschungsgemeinschaft (DFG) through the excellence cluster 'The Hamburg Center for Ultrafast Imaging (CUI) – Structure, Dynamics and Control of Matter at the Atomic Scale' (EXC1074) and the Joachim Herz Foundation.

## Additional information

### Funding

| Funder | Grant reference number | Author |
| --- | --- | --- |
| Marie Curie Actions | PCOFUND-GA-2008-229597 | Arjen J Jakobi |
| Marie-Sklodowska-Curie | PIEF-GA-2012-331285 | Arjen J Jakobi |
| Deutsche Forschungsge-meinschaft | Hamburg Center for Ultrafast Imaging (CUI) | Arjen J Jakobi |
| Joachim Herz Stiftung | | Arjen J Jakobi |

The funders had no role in study design, data collection and interpretation, or the decision to submit the work for publication.

## Author contributions
Arjen J Jakobi, Conceptualization, Resources, Data curation, Software, Formal analysis, Validation, Investigation, Visualization, Methodology, Writing—original draft, Writing—review and editing; Matthias Wilmanns, Conceptualization, Supervision, Methodology, Writing—original draft, Project administration; Carsten Sachse, Conceptualization, Resources, Software, Supervision, Investigation, Visualization, Methodology, Writing—original draft, Project administration, Writing—review and editing

## Author ORCIDs
Arjen J Jakobi (iD) http://orcid.org/0000-0002-7761-2027
Carsten Sachse (iD) http://orcid.org/0000-0002-1168-5143

## Decision letter and Author response
Decision letter https://doi.org/10.7554/eLife.27131.035
Author response https://doi.org/10.7554/eLife.27131.036

# Additional files

## Supplementary files
• Transparent reporting form
DOI: https://doi.org/10.7554/eLife.27131.022

## Major datasets
The following previously published datasets were used:

| Author(s) | Year | Dataset title | Dataset URL | Database, license, and accessibility information |
|---|---|---|---|---|
| Hoffmann NA, Jakobi AJ, Moreno-Morcillo M, Glatt S, Kosinski J, Hagen WJ, Sachse C, Muller CW | 2015 | Cryo-EM structure of yeast RNA polymerase III at 4.7 A | http://www.ebi.ac.uk/pdbe/entry/emdb/EMD-3180 | Publicly available at the EMDataBank (accession no: EMD-3180) |
| Liao M, Cao E, Julius D, Cheng Y | 2013 | Structure of the capsaicin receptor, TRPV1, determined by single particle electron cryo-microscopy | http://www.ebi.ac.uk/pdbe/entry/emdb/EMD-5778 | Publicly available at the EMDataBank (accession no: EMD-5778) |
| Bai XC, Yan CY, Yang GH, Lu PL, Ma D, Sun LF, Zhou R, Scheres SHW, Shi YG | 2015 | Cryo-EM structure of the human gamma-secretase complex at 3.4 angstrom resolution. | http://www.ebi.ac.uk/pdbe/entry/emdb/EMD-3061 | Publicly available at the EMDataBank (accession no: EMD-3061) |
| Bartesaghi A, Merk A, Banerjee S, Matthies D, Wu X, Milne JL, Subramaniam S | 2015 | 2.2 A resolution cryo-EM structure of beta-galactosidase in complex with a cell-permeant inhibitor | http://www.ebi.ac.uk/pdbe/entry/emdb/EMD-2984 | Publicly available at the EMDataBank (accession no: EMD-2984) |
| Fischer N, Neumann P, Konevega AL, Bock LV, Ficner R, Rodnina MV, Stark H | 2015 | 2.9A structure of E. coli ribosome-EF-Tu complex by Cs-corrected cryo-EM | http://www.ebi.ac.uk/pdbe/entry/emdb/EMD-2847 | Publicly available at the EMDataBank (accession no: EMDB-2847) |
| Campbell MG, Veesler D, Cheng A, Potter CS, Carragher B | 2015 | 2.8 Angstrom resolution reconstruction of the T20S proteasome | http://www.ebi.ac.uk/pdbe/entry/emdb/EMD-6287 | Publicly available at the EMDataBank (accession no: EMDB-6287) |

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
