## [Decision Letter]

Thank you for submitting your article "Iterative model-based density improvement yields better atomic structures from cryo-EM maps" for consideration by *eLife*. Your article has been favorably evaluated by John Kuriyan (Senior Editor) and three reviewers, one of whom is a member of our Board of Reviewing Editors. The following individual involved in review of your submission has agreed to reveal his identity: Tom Terwilliger (Reviewer #2).

The reviewers have discussed the reviews with one another and the Reviewing Editor. As you will see from the detailed review below, they have identified several issues that are sufficiently serious to preclude publication in *eLife*. Nevertheless, the editor and the reviewers feel that it might be possible, and also worthwhile, for you to address these issues in a revised manuscript. We emphasize, however, that the reservations are serious ones, and that a positive decision concerning publication can only be made if the revised manuscript satisfactorily addresses these concerns.

Summary:

The authors present a new contrast enhancement (or local sharpening) method to improve the appearance of EM maps using local rather than global amplitude scaling against an atomic reference structure. Considering the substantial recent advances that have been made in electron microscopy instrumentation and data analysis, the development of enhancement methods for EM density map is a timely topic.

Specifically, optimal "sharpening" of these maps may vary from location to location in the map. The authors present a method using correlation to a map calculated from an atomic model to identify an optimal sharpening for each location in the map and to systematically apply this local sharpening.

The paper could be substantially strengthened however, if the authors could address the fundamental concerns that the reviewers had.

Essential revisions:

1) In Figure 3 it is shown that a randomization of the regions of the reference model seems to produce as good (or even better?) results as the actual reference model. Why not always use a randomized reference to prevent even the slightest possibility of bias? Along similar lines, could one simply use a random set of atoms for the local amplitude scaling procedure? Finally, other, non-model based, scaling may do just as well as the proposed method. This point merits further investigation.

2) Examining Figure 3, it appears that the side chain density is substantially less when the side chains are not included in the LocScale procedure. This could be due to at least two different effects. (1) The map is now being targeted to zero density at the side chains and therefore the sharpening will be poorer, or (2) the procedure is biased by the model. The argument that the method has no model bias is not supported very convincingly by this figure. It might be more convincing if just one side chain at a time were removed from the model and it could be seen that this side chain density comes back just as strongly as if it had been included. Perhaps a numerical assessment of side chain density could help as well. Additionally it would be helpful to run a test showing whether the presence of atoms in the model increases density in positions where there are not really atoms present (this is what model bias is really all about).

3) One of the reviewers had a look at one of the two maps (emd_5778) given as an example showing the power of the method. Figure 3 shows the map from the EMD; in the region shown (in particular residues 420-436) very little side-chain density is evident. Then after applying the LocScale method, more side chains can be seen. This reviewer downloaded the map and carried out the following operations using nightly-build versions of phenix available to anyone to sharpen the overall map (no local sharpening): phenix.map_box 3j5p.pdb emd_5778.map, phenix.auto_sharpen 3j5p_box.ccp4 resolution=4.2. The resulting map is highly interpretable and clearly shows side chains in the region of residues 420-436. Based on this, it is not at all clear that the LocScale approach has substantially improved the map beyond what can be accomplished by simple adjustment of overall Fourier coefficients without local sharpening.

4) Following up on the previous point, it is somewhat hard to tell from Figure 3, Figure 4, and Figure 5 that model-scaled maps are substantially better than the original maps (other than having background noise removed, of course). On that topic, what happens if a portion of a model is omitted? Does that part of the map disappear, leading to an apparent better agreement between model and map?

5) Contrast-enhanced (or locally sharpened) maps may be particularly useful for display purposes to achieve the same degree of sharpening in a single figure for a publication or for a lecture. During manual model building by a structural biologist, he/she can adjust the degree of sharpening interactively (for example, Coot has a nice feature to do this in real time). Conversely, when models are automatically built, local sharpening may be less important since the building method takes the entire 3D map into account rather than a particular isosurface representation, and local variation in apparent "resolution" are absorbed by the atomic B-factor of the model. A discussion along these lines would be desirable.

6) The authors do not appear to address the key question of how many parameters are effectively being used to adjust the density to match the model-based density. The authors use one scale factor per pixel, but the correlation is carried out over a larger region so the number of effective parameters is fortunately not as high as the number of grid points. Nevertheless, in the limiting case, it might be possible to adjust the low vs high-resolution terms affecting each grid point individually, and therefore to have a huge amount of model bias. The paper would be much stronger if this could be explicitly discussed, including more than the short description provided on how an appropriate window size can be chosen. The authors currently use 7 times the pixel spacing as the size of the window over which sharpening is carried out. This means that points only up to 3 pixels away are considered in sharpening. Pixel size is highly variable in cryo-EM maps so this may be poorly-defined, but a typical pixel size would be about 1/4 to 1/6 the effective resolution, meaning that points considered in the sharpening factors for one position would be only up to about 1/2 to 3/4 the resolution away (or a 2-3 A radius for a 4 A map). This would appear to have very high potential for model bias. Intuitively it would seem that the appropriate radius for sharpening should be about the distance over which the optimal sharpening changes substantially, which in a real map might normally be at least on the order of 10 A or more. The authors could strengthen the paper with both a theoretical discussion of this issue and a practical demonstration of how the results are affected by varying the box size.

7) The use of the analogy to "density modification" in crystallographic refinement is highly misleading and should be removed. The two situations are fundamentally different. In the X-ray diffraction case, information about the density in one physical location in the crystal can be transferred to another, separate, location in the crystal. For example the expected flatness of the solvent can be used to improve the density in the protein region. The reason that this can occur is that the amplitudes are measured accurately and the amplitudes contain information about the differences between the true and working electron density in the cell. In the cryo-EM case, no information is transferred from one place to another in this way. All that is happening is that the power of Fourier terms at varying resolutions is being changed. This is so different from density modification that the use of the terms and the analogy is unwarranted and misleading.

8) The local sharpening has some superficial resemblance to the "unsharp masking" method in the Feature Enhanced Map (FEM) calculation in phenix (Afonine, et al., Acta Crystallographica Section D 71, 646-666, 2015). The "unsharp masking" method takes an average of nearest density points to locally sharpen the map rather than scaling the map locally against a radially averaged reference model. So, the two algorithms are quite different, but I wonder if the net result is similar, i.e., a map that has the same level of "local sharpening" throughout the map. It would be interesting to compare the two approaches, but at the minimum, the authors should discuss the similarities and differences of the two methods.

9) It is not surprising that EMRinger and FSC (map versus model) scores improve – maps and models must get more similar when the map amplitudes are made more similar to the model amplitudes than they were. A phase-only comparison of the (adjusted) model versus map would be more informative. The Molprobity score improving is nice, but one can improve a molprobity score by artificially enforcing good model geometry. The real value of the technique will be if it can lead to features appearing in maps that are real, without those features being present in the input model. Figure 4 attempts to make that point, but I do not think it does so in a compelling way. Figure 4 probably does it best.

10) Perhaps it would be easier to assess the effect of the process from looking at actual maps rather than figures?

[Editors' note: further revisions were requested prior to acceptance, as described below.]

Thank you for submitting your article "Model-based local density sharpening improves atomic interpretation of cryo-EM maps" for consideration by *eLife*. Your article has been favorably evaluated by John Kuriyan (Senior Editor) and three reviewers, one of whom is a member of our Board of Reviewing Editors. The following individual involved in review of your submission has agreed to reveal his identity: Tom Terwilliger (Reviewer #2).

The reviewers have discussed the reviews with one another and the Reviewing Editor has drafted this decision to help you prepare a revised submission. Given the nature of this contribution, the Board and reviewers agree that we could consider this as a Tools and Resources paper but not as a Research Article.

We would like to thank the authors for addressing many of the concerns raised by the reviewers. However, the most important criticism of the paper, which is that the authors have not shown that the LocScale method is actually an improvement over simpler existing methods, has not been satisfactorily addressed. To the contrary, the new data that are presented give every indication that the LocScale method does not significantly improve maps beyond what is achievable with overall B-factor sharpening. Nonetheless, an alternative approach such as this is seen as a valuable contribution.

As requested in the previous review, the authors now included comparisons of the LocScale method with other methods, including an overall B-factor sharpening. The authors now show a portion of the LocScale map for emd_5778 that is similar to the one in the previous version of the paper, along with maps from these other methods that can be seen to be of very similar overall quality. The emd_5778 figures in Figure 2 show that the B-factor sharpened map is just as good as the LocScale map. Even in the helical region shown in Figure 2 the global B-factor map looks just as good as the LocScale map.

Throughout the paper, beginning with the title, the authors claim that LocScale improves density maps. It is certainly true that LocScale improves these maps relative to the deposited maps. However it is easy to demonstrate that these deposited maps are necessarily optimal and existing methods that carry out overall sharpening can also improve maps in many cases (as the authors have now done for emd_5778 for example, and as it is easy to show for emd_2984 as well). Therefore a claim of improving these maps is misleading and is not appropriate.

In order to claim that the LocScale method improves maps, the authors need to use current methods for sharpening as comparisons throughout the paper and demonstrate that LocScale consistently improves the maps beyond the capability of existing global sharpening methods. However, if improvements cannot be clearly documented relative to overall sharpening methods, claims of improvement should be removed throughout the manuscript, including the statement in the Abstract that your method reveals previously undiscovered structural detail.

---

## [Author Response]

Essential revisions:1) In Figure 3 it is shown that a randomization of the regions of the reference model seems to produce as good (or even better?) results as the actual reference model. Why not always use a randomized reference to prevent even the slightest possibility of bias? Along similar lines, could one simply use a random set of atoms for the local amplitude scaling procedure?

We realized that the term randomization of the reference model had the potential of being misinterpreted as it was not referred to as a true random reference window, but as stated in the manuscript: “…we used randomized reference windows with approximately equivalent resolution that originated from completely different regions of the reference map”. The term we use now to avoid this confusion is “structurally unrelated reference window”. This distinction is important and we make use of this example when explaining the effect of atomic B-factors and local resolution in more detail as part of the response to point 5.

Based on the principal suggestion of using randomization of coordinates, we have performed additional tests and included the following paragraph together with Figure 3—figure supplement 1 to the revised version of the manuscript:

“In order to further assess the potential of coordinate bias, we randomly perturbed atom positions of the model within a mask encompassing the molecular outline and computed reference maps from the perturbed models to scale the experimental density. For perturbations ranging from 2.5 – 50 Å r.m.s.d., side chain densities are readily observed despite the reference having no resemblance to the original structure. We do, however, observe a noticeable decrease in map contrast with increasing r.m.s.d (Figure 3—figure supplement 1).”

The suboptimal sharpening result obtained with strongly perturbed model coordinates is dealt with in our response to point 5.

Finally, other, non-model based, scaling may do just as well as the proposed method. This point merits further investigation.

Based on the suggestion of the referee, we have systematically compared the result of various global and local sharpening approaches such as global B-factor sharpening, local Guinier B-factor sharpening and unsharp masking. We introduce the topic by demonstrating the need for local sharpening approaches by estimating local Guinier B-factors (see Materials and methods for details; this option will also be part of the distributed program to permit model-free local sharpening).

Our comparison reveals that, in general, local sharpening approaches outperform global sharpening methods when resolution variation exists. While local sharpening methods generally improve map representations, we see additional benefit when using model-based sharpening as proposed. The following paragraphs including Figure 1 have been introduced into the manuscript, Figure 2; Figure 2—figure supplement 1, respectively:

“We quantified this amplitude decay and mapped locally determined B-factors to four deposited 3D densities from the EM databank (Figure 1, Figure 1—figure supplement 1). […] The results from our four test cases suggest that locally adjusted sharpening levels may be required to optimally represent density features, in particular for maps with significant local resolution variation.”

“In order to account for local differences in the estimated B-factor, we implemented three different local sharpening procedures and compared them with the commonly used global sharpening approach. […] In conclusion, reference-based sharpening by local amplitude scaling provides a robust way to sharpen and filter all regions in a map such that they conform to the expected amplitude falloff.”

2) Examining Figure 3, it appears that the side chain density is substantially less when the side chains are not included in the LocScale procedure. This could be due to at least two different effects. (1) The map is now being targeted to zero density at the side chains and therefore the sharpening will be poorer, or (2) the procedure is biased by the model. The argument that the method has no model bias is not supported very convincingly by this figure. It might be more convincing if just one side chain at a time were removed from the model and it could be seen that this side chain density comes back just as strongly as if it had been included. Perhaps a numerical assessment of side chain density could help as well. Additionally it would be helpful to run a test showing whether the presence of atoms in the model increases density in positions where there are not really atoms present (this is what model bias is really all about).

We thank the referee for his/her suggestion for improving the presentation for the assessment of model bias. We now included the following paragraph including Figure 3/D into the revised version of the manuscript:

“Atomic model coordinates do not introduce feature bias Model bias is commonly understood as the appearance or disappearance of map features not present in the experimental data, but imposed by the model. […] While side chain densities are slightly attenuated when compared at equivalent density threshold, they are readily interpretable despite the absence of any side chain information in the reference model.”

Based on the suggestion of the reviewer, we also computed a numerical assessment of the side chain density in the form of a real-space correlation coefficient (Author response image 1). The real-space correlation graphs of the wild-type model with the differently scaled LocScale maps are almost identical and confirm the visual perception of the scaled densities in Figure 3. Both data show that the density fit is not sensitive to point mutations in the reference and only slightly decrease the correlation with LocScale maps scaled with a truncated poly-alanine reference.

**Author response image 1. respfig1:** Quantification of reference model perturbation on LocScale maps. (**A**) Residue-averaged real space correlation (RSCC) for TRPV1 residues 430-449 computed between the original PDB (WT) and LocScale maps obtained with scaling using a WT, F436A, V440W or poly-Ala reference map. (**B**) Residue-averaged real space correlation (RSCC) for TRPV1 residues 430-449 computed between the original PDB (WT) and the simulated reference maps obtained from WT, F436A, V440W or poly-Ala coordinate models.

3) One of the reviewers had a look at one of the two maps (emd_5778) given as an example showing the power of the method. Figure 3 shows the map from the EMD; in the region shown (in particular residues 420-436) very little side-chain density is evident. Then after applying the LocScale method, more side chains can be seen. This reviewer downloaded the map and carried out the following operations using nightly-build versions of phenix available to anyone to sharpen the overall map (no local sharpening): phenix.map_box 3j5p.pdb emd_5778.map, phenix.auto_sharpen 3j5p_box.ccp4 resolution=4.2. The resulting map is highly interpretable and clearly shows side chains in the region of residues 420-436. Based on this, it is not at all clear that the LocScale approach has substantially improved the map beyond what can be accomplished by simple adjustment of overall Fourier coefficients without local sharpening.

We appreciate the criticism by the reviewer as we did not intend to suggest that global sharpening methods cannot improve the density of the particular region of residues 420-436 in EMD-5778. As stated in the response to point 1, we have now compared alternative sharpening approaches including global sharpening as implemented in Phenix’ auto_sharpen. Phenix auto_sharpen correctly estimates the average global B-factor to approximately -100 Å2. As shown in Figure 1 this is close to the estimated local B-factor for the residues 420449 in Figure 3. As a result, we agree with the reviewer that a map sharpened with a global B-factor of -100 Å2 yields comparable detail for the highlighted region. As Figure 1 illustrates, however, even for TRPV1 there is considerable variation in the local B-factors estimated from local Guinier analysis, suggesting that local sharpening is warranted also for the example of TRPV1. Our comparison of different local sharpening methods (point 1, second response) with global sharpening confirm this picture and suggests that LocScale sharpening has additional benefit over global methods in the presented case.

4) Following up on the previous point, it is somewhat hard to tell from Figure 3, Figure 4, and Figure 5 that model-scaled maps are substantially better than the original maps (other than having background noise removed, of course).

We believe to have shown in detail that density improvement can be obtained with local sharpening throughout the section “LocScale maps facilitate model building and reveal novel structural detail”. We show how LocScale maps can aid manual model building – a process guided by visual perception of the displayed densities. At certain resolutions noise and true map features are not easily distinguishable. For example, Figure 4 reveals that some water molecules have been placed correctly whereas some other water molecules may have inadvertently been placed into noise peaks. This example showcases that down-weighting noise and sharpening true high-resolution features appropriately are not limited to display purposes. The same is true for the structures treated in Figure 5 and 6. We agree with the referee’s point 10 that inspection of the 3D maps is more useful than selected 2D presentations. Therefore, we provide the LocScale sharpened maps in the process of revision.

On that topic, what happens if a portion of a model is omitted? Does that part of the map disappear, leading to an apparent better agreement between model and map?

We are thankful to the reviewer for raising this important point as it has not been dealt with in appropriate detail. For clarity, we add the following paragraph and Figure 4 to the revised manuscript:

“One of the potential shortcomings of the proposed method is the principal requirement of an atomic model. […] This model served as a reference for additional cycles of model building, refinement and sharpening until model completion.”

5) Contrast-enhanced (or locally sharpened) maps may be particularly useful for display purposes to achieve the same degree of sharpening in a single figure for a publication or for a lecture.

We agree and therefore we provided visual examples where the improvement in density display is supported. As outlined in our first response to point 4, we nevertheless are convinced that the benefit for such densities goes beyond mere visualization purposes. We demonstrated the benefits for the building of atomic models and for model refinement in the sections: “LocScale maps facilitate model building and reveal novel structural detail” and “LocScale maps are improved map targets for model refinement”, respectively.

During manual model building by a structural biologist, he/she can adjust the degree of sharpening interactively (for example, Coot has a nice feature to do this in real time).

Although interactive sharpening is indeed a very useful tool, the subjective decision on how to sharpen the map locally will often require prior model building experience of the user. One of the motivations of our approach was to reduce the subjectivity involved in interactive thresholding and sharpening. While interactive sharpening and thresholding are and will remain important tools for model building, the property of LocScale maps to represent most parts of the map at the appropriate detail simultaneously at a single threshold level does represent an improvement over current procedures and this way will further facilitate the process of manual model building.

Conversely, when models are automatically built, local sharpening may be less important since the building method takes the entire 3D map into account rather than a particular isosurface representation, and local variation in apparent "resolution" are absorbed by the atomic B-factor of the model. A discussion along these lines would be desirable.

In the present study, we have primarily focused on benefits of locally sharpened maps for map interpretation by manual model building and we acknowledge that we have not systematically tested the relevance for variable map sharpening for automated model building procedures. We are currently aware of only one study in which automated model building has been systematically compared for sharpened and unsharpened maps. Liu et al. (Liu, C. and Xiong, Y., J Mol Biol 426, 980–993 (2014)) compared 19 structures for which atomic models were obtained with various automatic model building procedures using unmodified, sharpened and anisotropically sharpened electron density maps. The authors found an improvement in the built models for the majority of cases when sharpened maps were used in the procedure. These results suggest that there may be benefit for local sharpening also for automated model building procedures using cryo-EM density maps. We note that the results obtained by Liu et al. refer to improvement of crystallographic models after uniform sharpening by an automatically determined or manually fitted B-factor. We further note that the resolution of the majority of models used in their study exceeds that of those used in the present study. Whether the LocScale procedure, or local sharpening by other means, can benefit automated model building into cryo-EM maps will be an interesting subject for future studies.

Although we mentioned the importance of the atomic B-factors in the Discussion: “Once estimates of atomic B-factors have been obtained, LocScale maps provide locally optimized contrast and facilitate further rounds of model building and refinement”, we may have understated its relative importance. Therefore, we dedicate an entire section under the subheading: “Atomic B-factors are required for optimal sharpening” to this topic.

“Atomic B-factors are required for optimal sharpening. Coordinate perturbations of the reference did not result in appearance or disappearance of features in the sharpened map, whereas coordinate randomization beyond a certain r.m.s.d. resulted in suboptimal sharpening. […] This example confirms our previous conclusion that a structurally unrelated window can be used for scaling without introducing features from the model, while it also illustrates how the result of sharpening is dependent on the correct estimation of the slope of the amplitude falloff in the reference window, which in the LocScale procedure is determined from the refined atomic B-factors of the atomic model.”

6) The authors do not appear to address the key question of how many parameters are effectively being used to adjust the density to match the model-based density. The authors use one scale factor per pixel, but the correlation is carried out over a larger region so the number of effective parameters is fortunately not as high as the number of grid points. Nevertheless, in the limiting case, it might be possible to adjust the low vs high-resolution terms affecting each grid point individually, and therefore to have a huge amount of model bias. The paper would be much stronger if this could be explicitly discussed, including more than the short description provided on how an appropriate window size can be chosen. The authors currently use 7 times the pixel spacing as the size of the window over which sharpening is carried out. This means that points only up to 3 pixels away are considered in sharpening. Pixel size is highly variable in cryo-EM maps so this may be poorly-defined, but a typical pixel size would be about 1/4 to 1/6 the effective resolution, meaning that points considered in the sharpening factors for one position would be only up to about 1/2 to 3/4 the resolution away (or a 2-3 A radius for a 4 A map). This would appear to have very high potential for model bias. Intuitively it would seem that the appropriate radius for sharpening should be about the distance over which the optimal sharpening changes substantially, which in a real map might normally be at least on the order of 10 A or more. The authors could strengthen the paper with both a theoretical discussion of this issue and a practical demonstration of how the results are affected by varying the box size.

We thank the referee for bringing up the point of the number of effective parameters. We added an entire paragraph in the Discussion section of the revised manuscript:

a) Theoretical discussion:

“Due to the high number of voxels present in a 3D EM density map, our procedure requires a large number of parameters and poses a risk of overfitting. […] Consequently, the two fitted parameters are kept locally restrained, thus reducing the risk for overfitting of local sharpening profiles.”

b) Practical tests on window size:

We thank the referee for bringing up the point of window size of the procedure. The referee’s concern results partially from a misunderstanding of the method section, that we realize was not phrased unambiguously. We stated: “As a typical window size parameter, we used seven times the resolution in pixels.” In this case, we do not refer to the resolution at the Nyquist limit, but to the effective (average) resolution of the density map in Å, expressed in pixels. We apologize for the unclear phrasing, but emphasize that the smallest window size used was 17 Å (26 px) for the 2.2 Å resolution β-galactosidase map. Specifically the window sizes used for the LocScale maps shown in Figure 2, and Figure 2—figure supplement 2 were 24.3 Å (20 px) for TRPV1, 23.8 Å (17 px) for γ-secretase and 33.6 Å (31 px) for RNA Pol III. We do concur with the reviewer that the correlation distances for atomic B-factors will be at least in the range mentioned. Our detailed analysis for the TRPV1 case (Figure 3—figure supplement 3) suggest that the choses window size (24 Å) in this case was close to optimal for maximizing the sharpening level across the map. More precisely we state in the revised Materials and methods section of the manuscript:

“As a typical window size parameter, we used seven times the effective resolution (FSC 0.143 cutoff) in pixels in an effort to balance small window size to preserve locality of the approach against large window size to have sufficient number of Fourier pixels to appropriately sample the fine-structure of the amplitude fall-off.”

In addition, we systematically tested the effect of different window sizes and include the results and add the entire Figure 3—figure supplement 3 to address this issue:

“The effect of window size on the scaled densities. The LocScale procedure employs scaling of radial amplitude profiles that are averaged over a rolling density window spanning dimensions between 15 and 45 Å in the described examples. […] Our results indicate that an optimal window size can be chosen that best reflects the effective distance of local B-factor correlations, thus maximizing map kurtosis and Debye effect contrast across the sharpened map.**”**

7) The use of the analogy to "density modification" in crystallographic refinement is highly misleading and should be removed. The two situations are fundamentally different. In the X-ray diffraction case, information about the density in one physical location in the crystal can be transferred to another, separate, location in the crystal. For example the expected flatness of the solvent can be used to improve the density in the protein region. The reason that this can occur is that the amplitudes are measured accurately and the amplitudes contain information about the differences between the true and working electron density in the cell. In the cryo-EM case, no information is transferred from one place to another in this way. All that is happening is that the power of Fourier terms at varying resolutions is being changed. This is so different from density modification that the use of the terms and the analogy is unwarranted and misleading.

Based on the suggestion of the referee, we removed the analogy to the term density modification. We have also removed introductory statements on the topic of density modification to avoid any confusion.

8) The local sharpening has some superficial resemblance to the "unsharp masking" method in the Feature Enhanced Map (FEM) calculation in phenix (Afonine, et al., Acta Crystallographica Section D 71, 646-666, 2015). The "unsharp masking" method takes an average of nearest density points to locally sharpen the map rather than scaling the map locally against a radially averaged reference model. So, the two algorithms are quite different, but I wonder if the net result is similar, i.e., a map that has the same level of "local sharpening" throughout the map. It would be interesting to compare the two approaches, but at the minimum, the authors should discuss the similarities and differences of the two methods.

We treat the unsharp masking method in our response to point 1.

9) It is not surprising that EMRinger and FSC (map versus model) scores improve – maps and models must get more similar when the map amplitudes are made more similar to the model amplitudes than they were.

We share the referee’s principal concern and realized that we have not clearly stated the context of the correlation score in the main text, although it was done in the Materials and methods section: “All cross-validation FSC and real-space correlations were computed against the original reconstruction and the deposited EMDB map, respectively”, as well as in the legend to Table 1 “Overall real-space correlation computed at the average map resolution using a soft mask around atoms. The EMDB deposition was used as the reference map in all cases” and in the legend to Figure 6—figure supplement 1 “All FSC_ref_ curves were computed using the original density map”. We re-emphasized explaining our similarity score in the main text:

“In every case, the overall real-space correlation improved by 3 – 6% for models refined against the LocScale map when measured against the deposited EMDB entry (Table 1), and FSC curves computed between model maps and the original reconstructions show small but notable improvements of the LocScale model fit (Figure 6—figure supplement 1). It should be noted that comparisons of real-space correlation and model-map FSC measures are only meaningful when they are calculated against a common reference volume that has not undergone model-based scaling.”

For clarification: in the original maps used for model vs. map FSC computation, there was no information transferred from model amplitudes. Our results clearly suggest that the models refined against LocScale are more consistent with the original data than models refined against globally sharpened or locally filtered maps. EMRinger scores have been reported by the original authors to be sensitive to sharpening, but generally benefit from small amounts of sharpening for maps at lower resolution. We attribute the improved scores of LocScale maps to the fact that local sharpening or blurring more accurately represents the density and leads to better rotameric assignment. We do also note that the observed improvements are small. We agree that Molprobity scores can be misleading when geometry was artificially restrained.

The parametrization of the refinement, however, was identical for the respective maps among the individual structures tested and LocScale maps used as refinement target appear to slightly improve these scores for most cases. The worsening of scores and the increase in occurrence of outliers in other cases suggest that there may not be only benefit, or alternatively, that those deviations are possibly real. In addition to the combination of improved scores, we also provide data for the better agreement of model vs. map FSCs after refinements based on the LocScale maps, which are independent to amplitude scaling differences as they are calculated within resolution shells.

A phase-only comparison of the (adjusted) model versus map would be more informative. The Molprobity score improving is nice, but one can improve a molprobity score by artificially enforcing good model geometry. The real value of the technique will be if it can lead to features appearing in maps that are real, without those features being present in the input model.

Based on the suggestion of the referee, we also computed differential phase residuals of the refined model maps against the original reconstruction. The curves are summarized in Author response image 2. The phase-only comparison shows a slight but consistent improvement of agreement over the refinement structures from the globally sharpened maps and thus confirms the presented FSC measurements.

**Author response image 2. respfig2:** 

Figure 4 attempts to make that point, but I do not think it does so in a compelling way. Figure 4 probably does it best.

Based on the suggestion that former Figure 4 was not as convincing as Figure 4/E, we moved the Figure 4 to the current Figure 5—figure supplement 1.

10) Perhaps it would be easier to assess the effect of the process from looking at actual maps rather than figures?

We uploaded the LocScale sharpened maps for visual inspections by the referees.

[Editors' note: further revisions were requested prior to acceptance, as described below.]

We would like to thank the authors for addressing many of the concerns raised by the reviewers. However, the most important criticism of the paper, which is that the authors have not shown that the LocScale method is actually an improvement over simpler existing methods, has not been satisfactorily addressed. To the contrary, the new data that are presented give every indication that the LocScale method does not significantly improve maps beyond what is achievable with overall B-factor sharpening. Nonetheless, an alternative approach such as this is seen as a valuable contribution.

We appreciate that the reviewers consider the presented approach a valuable alternative contribution to currently most widely used B-factor sharpening for contrast improvement of cryo-EM maps. We have adapted our manuscript to reflect the raised criticism of the reviewers that the proposed method does not significantly improve the quality of the maps beyond that of overall sharpened maps in cases of relatively homogeneous resolution distribution. We still point out that local sharpening can provide benefits over global sharpening in cases in particular when differences in local resolution exist. To what extent these benefits are significant and can be called improvements is indeed difficult to define. We acknowledge this now by more careful wording. We do not claim that the method universally improves maps. In maps with large variation in local resolution, model building does require differently sharpened maps and interactive thresholding. We do believe that locally sharpened maps have useful properties that make the handling of multiple maps obsolete, reduce subjectivity and thus facilitate manual, and possibly automatic, model building.

As requested in the previous review, the authors now included comparisons of the LocScale method with other methods, including an overall B-factor sharpening. The authors now show a portion of the LocScale map for emd_5778 that is similar to the one in the previous version of the paper, along with maps from these other methods that can be seen to be of very similar overall quality. The emd_5778 figures in Figure 2 show that the B-factor sharpened map is just as good as the LocScale map. Even in the helical region shown in Figure 2 the global B-factor map looks just as good as the LocScale map.

We acknowledge the perception of the reviewers that overall emd_5778 presentations (Figure 2) do not easily reveal the benefits of the LocScale approach. In particular, the presented transmembrane helix of emd_5778 (Figure 2) is an example for which the benefit of local sharpening is limited since the estimated local sharpening level coincides with the estimated overall B-factor for this map (Figure 1). This comparison was intentionally included as it illustrates how the approach works. We had, however, included the following support highlighting the differences between sharpening approaches in the revised manuscript:

· Figure 2 and Figure 6 (now Figure 7): LocScale densities showing clearer detail of helical pitch and side chain densities in peripheral density regions and more continuous density in loop regions;

· Figure 1: local B-factor estimation using Guinier analysis (Figure 1) shows that emd_5778 map does not have a single B-factor as it is applied in overall B-factor sharpening;

· We provided LocScale maps for reviewers for detailed 3D inspection as 2D illustrations of selected density fragments may not always allow straightforward illustration and judgement of these differences;

· Map kurtosis is maximized in LocScale maps.

We furthermore find as a result of the procedure that LocScale maps represent detail throughout the map when visualized at a single threshold level. Using common overall sharpening methods, however, multiple differently sharpened and thresholded maps are necessary to achieve this. We illustrate this useful property more clearly in the following new paragraph and new Figure 6, Figure 6—figure supplement 1, Figure 6—figure supplement 2:

“One notable benefit of using locally scaled maps is that all parts of the map can be simultaneously visualized at a single threshold level. […] Similar observations can be made for peripheral densities in the 20S proteasome structure that are substantially weaker than the better resolved core structure (Figure 6—figure supplement 2).”

Together, our view is that any contrast enhancement reducing ambiguity of the modelled conformation provides benefits to map interpretation.

Throughout the paper, beginning with the title, the authors claim that LocScale improves density maps.

Based on the criticism we shortened the title to a descriptive statement more compatible with the Resources and Tools format of *eLife*: “Model-based local density sharpening of cryo-EM maps”.

It is certainly true that LocScale improves these maps relative to the deposited maps. However it is easy to demonstrate that these deposited maps are necessarily optimal and existing methods that carry out overall sharpening can also improve maps in many cases (as the authors have now done for emd_5778 for example, and as it is easy to show for emd_2984 as well). Therefore a claim of improving these maps is misleading and is not appropriate.

We agree with the reviewers that two of the presented maps (emd_5778 and emd_2984) have not been optimally sharpened in the original depositions. In these cases, certain improvements can also be achieved by the other methods, in particular when adjusted locally. For this reason, we had included in the previous submission:

· Sharpening adjustments to emd_5778 (Materials and methods: “The deposited map for TRPV1 (EMD-5778) was globally sharpened using an additional B-factor of -100 Å2.”)

To better illustrate the fact that other sharpening methods can achieve similar results when adjusted for particular map regions, we added in this revised version of the manuscript:

· Figure 6 (now 7A): emd_5778 density is presented after additional sharpening;

· Figure 5—figure supplement 1: emd_2984 showing the described loop region after additional blurring.

Although this procedure of overall sharpening or blurring certainly improve visibility of the displayed side chains or loop regions, it concomitantly does lead to diminished feature representation in other regions of the map. These types of operations can be readily achieved by on-the-fly blurring/sharpening using Coot. Locally sharpened maps such as the LocScale maps of the presented example do not require these additional adjustments and as such perhaps help reduce the level of subjectivity in map interpretation. To acknowledge the criticism, we have modified the title, the Abstract and the text to remove the term “density improvement” and refer to “facilitated interpretation” instead where appropriate.

In order to claim that the LocScale method improves maps, the authors need to use current methods for sharpening as comparisons throughout the paper and demonstrate that LocScale consistently improves the maps beyond the capability of existing global sharpening methods.

In the previous submission, we had included:

· Figure 2 and Figure 2—figure supplement 2: comparison of overall, local Guinier-based B-factor sharpening and UNSHARP masking with LocScale maps for all of the cited examples;

· Figure 3, Figure 5 (for emd_3180) and Figure 6 (now 7D-F) (emd_3061): deposited EMDB maps were sharpened using the currently most commonly applied and accepted practice of overall B-factor sharpening implemented in RELION. Note, in these cases overall sharpening is not able to reveal a comparable level of detail revealed in central slices through the map (Figure 2—figure supplement 2) and also is not able to confidently reveal additional glycosylation sites as observed in the LocScale map (Figure 5; Figure 2—figure supplement 2);

· Table 1: presented examples in the section on model refinement all had been compared with currently widely applied and accepted standards for overall sharpening.

To improve the illustration and better highlight the existing differences between the approaches in Figure 2—figure supplement 2 more obviously, we complemented this figure with additional pointers.

Our intention is not to discredit or disfavor existing methods, but to demonstrate how local sharpening can be used as a tool to facilitate interpretation of cryo-EM maps for the common case in which some degree of resolution variation is apparent. With this revision, we added Figure 5—figure supplement 1 for more details on comparative sharpening approaches and we also provide additional examples showcasing the potential of the method for *E. coli* ribosome EF-Tu complex, RNA Pol III and 20S proteasome (Figure 6 and Figure 6—figure supplement 1/2). In summary, we believe that our presented data and the new examples do support our proposal that LocScale maps can aid, or at least facilitate, model building.

However, if improvements cannot be clearly documented relative to overall sharpening methods, claims of improvement should be removed throughout the manuscript, including the statement in the Abstract that your method reveals previously undiscovered structural detail.

As requested and stated above, we have removed the terms “improvement” and “undiscovered structural detail” from the Abstract:

“Atomic models based on high-resolution density maps are the ultimate result of the cryo-EM structure determination process. […] By testing the procedure using six cryo-EM structures of TRPV1, β-galactosidase, γ-secretase, ribosome-EF-Tu complex, 20S proteasome and RNA polymerase III, we illustrate how local sharpening can increase interpretability of density maps in particular in cases of resolution variation and facilitates model building and atomic model refinement.”